# PARETO MANIFOLD LEARNING: TACKLING MULTIPLE TASKS VIA ENSEMBLES OF SINGLE-TASK MODELS

## ABSTRACT

In Multi-Task Learning, tasks may compete and limit the performance achieved on each other rather than guiding the optimization trajectory to a common solution, superior to its single-task counterparts. There is often not a single solution that is optimal for all tasks, leading practitioners to balance tradeoffs between tasks' performance, and to resort to optimality in the Pareto sense. Current Multi-Task Learning methodologies either completely neglect this aspect of functional diversity, and produce one solution in the Pareto Front predefined by their optimization schemes, or produce diverse but discrete solutions, each requiring a separate training run. In this paper, we conjecture that there exist Pareto Subspaces, i.e., weight subspaces where multiple optimal functional solutions lie. We propose *Pareto Manifold Learning*, an ensembling method in weight space that is able to discover such a parameterization and produces a continuous Pareto Front in a single training run, allowing practitioners to modulate the performance on each task during inference on the fly. We validate the proposed method on a diverse set of multi-task learning benchmarks, ranging from image classification to tabular datasets and scene understanding, and show that *Pareto Manifold Learning* outperforms state-of-the-art algorithms.

[All reviewers]: We have added "notes" such as this one to make the changes more visible. For minor things such as slightly changing plot size, we do not write anything.

## 1 INTRODUCTION

In Multi-Task Learning (MTL), multiple tasks are learned concurrently within a single model, striving towards infusing inductive bias that will help outperform the single-task baselines. Apart from the promise of superior performance and some theoretical benefits (Ruder, 2017), such as generalization properties for the learned representation, modeling multiple tasks jointly has practical benefits as well, e.g., lower inference times and memory requirements. However, building machine learning models presents a multifaceted host of decisions for multiple and often competing objectives, such as model complexity, runtime and generalization. Conflicts arise since optimizing for one metric often leads to the deterioration of other(s). A single solution satisfying optimally all objectives rarely exists and practitioners must balance the inherent trade-offs.

In contrary to single-task learning, where one metric governs the comparison between methods (e.g., top-1 accuracy in `ImageNet`), multiple models can be optimal in Multi-Task Learning; e.g., model X yields superior performance on task $\mathcal{A}$ compared to model Y, but the reverse holds true for task $\mathcal{B}$; thus, there is not a single better model among the two. This notion of tradeoffs is formally defined as *Pareto optimality*. Intuitively, improvement on an individual task performance can come only at the expense of another task. However, there exists no framework addressing the need for efficient construction of the Pareto Front, i.e., the set of all Pareto optimal solutions.

Recent methods in Multi-Task Learning casted the problem in the lens of multi-objective optimization and introduced the concept of Pareto optimality, resulting in different mechanisms for computing the descent direction for the shared parameters. Specifically, Sener & Koltun (2018) produce a single solution that lies on the Pareto Front. As an optimization scheme, however, it is biased towards the task with the smallest gradient magnitude, as argued in Liu et al. (2020). Lin et al. (2019) expand this idea and, by imposing additional constraints on the objective space to produce multiple solutions on the Pareto Front, each corresponding to a different user-specified tradeoff. Finally, the work by Ma et al. (2020) proposes an orthogonal approach that can be applied after training and starts with a discrete solution set and produces a continuous set (in weight space) around each so-

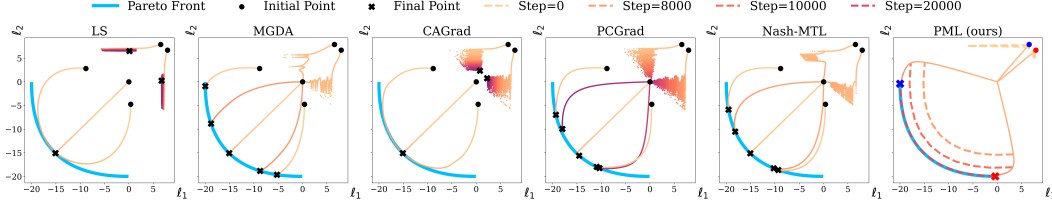

Figure 1: Illustrative example following Yu et al. (2020); Navon et al. (2022). We present the optimization trajectories in loss space starting from different initializations (black bullets) leading to final points (crosses). Color reflects the iteration number when the corresponding value is achieved. To highlight that our method (PML) deals in pairs of models, we use blue and red to differentiate them. Dashed lines show intermediate results of the discovered subspace. While baselines may not reach the Pareto Front or display bias towards specific solutions, PML discovers the entire Pareto Front *in a single run* and shows superior functional diversity.

lution, while the overall Pareto Front is continuous *only in objective space* as the union of the local (weight-space continuous) Pareto Fronts. As a consequence, the memory requirements grow linearly with the number of models stored. Navon et al. (2021); Lin et al. (2021) use hypernetworks to produce a Pareto Front in a single training run, but this approach has limited scalability and introduces additional design choices.

In this paper, we conjecture that we can actually produce a subspace with multiple Pareto stationary points in the Multi-Task Learning setting with the hypothesis that local optima (produced by different runs or sharing training steps) can be found in close proximity and are connected by simple paths. This is motivated by the recent advancements in single task machine learning that have explored the geometry of the loss landscape and shown experimentally that local optima are connected by simple paths, even linear ones in some cases (Wortsman et al., 2021; Garipov et al., 2018; Frankle et al., 2020; Draxler et al., 2018). We assume that, when the problem has multiple objectives, it acquires a new dimension relating to the number of tasks. Concretely, there are multiple loss landscapes and a solution that satisfies users' performance requirements must lie in the intersection of low loss valleys (for all tasks).

Building upon our conjecture, we develop a novel method, *Pareto Manifold Learning*, which casts Multi-Task problems as learning an ensemble of single-task predictors by interpolating among (ensemble) members during training. By operating in the convex hull of the members' weight space, each single-task model infuses and benefits from representational knowledge to and from the other members. During training, the losses are weighted in tandem with the interpolation, i.e., a monotonic relationship is imposed between the degree of a single-task predictor participation and the weight of the corresponding task loss. Consequently, the ensemble as a whole engenders a (weight) subspace that explicitly encodes tradeoffs and results in a continuous parameterization of the Pareto Front. We identify challenges in guiding the ensemble to such subspaces, designated *Pareto subspaces*, and propose solutions regarding balancing the loss contributions, and regularizing the Pareto properties of the subspaces and adapting the interpolation sampling distribution.

Experimental results validate that the proposed method is able to discover *Pareto Subspaces*, and outperforms baselines on multiple benchmarks. Our training scheme offers two main advantages. First, enforcing low loss for all tasks on a linear subspace implicitly penalizes curvature, which has been linked to generalization (Chaudhari et al., 2017), benefitting all tasks' performance. Second, the algorithm produces a subspace of Pareto Optimal solutions, rather than a single model, enabling practitioners to handpick during inference the solution that offers the tradeoff that best suits their needs.

[All reviewers]: Added a *short* clarification about related work. More details are also provided in "Related work" section.

## 2 RELATED WORK

**Multi-Task Learning.** Learning multiple tasks in the Deep Learning setting (Ruder, 2017; Crawshaw, 2020) is usually approached by architectural methodologies (Misra et al., 2016; Ruder et al., 2019), where the architectural modules are combined in several layers to govern the joint representation learning, or optimization approaches (Cipolla et al., 2018; Chen et al., 2018), where the architecture is standardized to be an encoder-decoder(s), for learning the joint and task-specific rep-

resentations, respectively, and the focus shifts to the descent direction for the shared parameters. We focus on the more general track of optimization methodologies fixing the architectural structure to Shared-Bottom (Caruana, 1997). The various approaches focus on finding a suitable descent direction for the shared parameters. The optimization methods can be broadly categorized into *loss-balancing* and *gradient-balancing* (Liu et al., 2020). For the former, the goal is to compute an appropriate weighting scheme for the losses, e.g., the losses can be weighted via task-dependent homoscedastic uncertainty (Cipolla et al., 2018), by enforcing task gradient magnitudes to have close norms (Chen et al., 2018). The latter class of methodologies manipulate the gradients so that they satisfy certain conditions; projecting the gradient of a (random) task on the normal plane of another so that gradient conflict is avoided (Yu et al., 2020), enforcing the common descent direction to have equal projections for all task gradients (Liu et al., 2020), casting the gradient combination as a bargaining game (Navon et al., 2022).

[All reviewers]: Removed discussion about Sener & Koltun (2018) from this point, since the paper is also discussed in the introduction and the next paragraph.

**Multi-Task Learning for Pareto Optimality.** The authors in (Sener & Koltun, 2018) were the first to view the search for a common descent direction under the Pareto optimality prism and employ the Multiple Gradient Descent Algorithm (MGDA) (Désidéri, 2012) in the Deep Learning context. However. MGDA did not account for task preferences and the solutions yielded for various initializations in a synthetic example resulted in similar points in the Pareto Front (Lin et al., 2019). By solving a slightly different formulation of the multi-objective problem, they are able to systematically introduce task trade-offs and produce a *discrete* Pareto Front. However, this approach requires as many training runs as the stated preference combinations and the optimization process for each training step of each run introduces a non-negligible overhead. The work in (Ma et al., 2020) proposes an orthogonal approach for Pareto stationary points; after a model is fitted with any Multi-Task Learning method and has converged to a point (seed) in parameter space, a separate phase seeks other Pareto stationary points in the vicinity of the seed. The convex hull of these points is guaranteed to lie in the Pareto Front. But training still needs to occur for every seed point, the separate phase overhead grows linearly with the number of additional models, and the Pareto Front is not continuous across seed points in *parameter space*. Navon et al. (2021) and Lin et al. (2021) employ hypernetworks to continuously approximate the Pareto Front in a single run, which introduces additional design choices. Ruchte & Grabocka (2021) address the scalability issues of hypernetworks by augmenting the feature space with the preference vector. Raychaudhuri et al. (2022) employ a second hypernetwork to also modulate the architecture of the target network addressing.

[All reviewers]: Added prior work.

**Ensemble Learning and Mode Connectivity.** Apart from Multi-Task Learning, our algorithm is methodologically tied to prior work in the geometry of the neural network optimization landscapes. The authors in (Garipov et al., 2018; Draxler et al., 2018) independently and concurrently showed that for two local optima $\boldsymbol{\theta}_1^*, \boldsymbol{\theta}_2^*$ produced by separate training runs (but same initializations) there exist nonlinear paths, defined as *connectors* by Wortsman et al. (2021), where the loss remains low. The connectivity paths can be extended to include linear in the case of the training runs sharing some part of the optimization trajectory (Frankle et al., 2020). These findings can be leveraged to train a neural network subspace by enforcing linear connectivity among the subspace endpoints (Wortsman et al., 2021). Appendix J discusses more related work regarding ensemble learning and flat minima.

## 3 PROBLEM FORMULATION

**Notation.** We use bold font for vectors $\boldsymbol{x}$, capital bold for matrices $\boldsymbol{X}$ and regular font for scalars $x$. $T$ is the number of tasks and $m$ is the number of ensemble members. Each task $t \in [T]$ has a loss $\mathcal{L}_t$. The overall multi-task loss is $\boldsymbol{L} = [\mathcal{L}_1, \ldots, \mathcal{L}_T]^\top$. $\boldsymbol{w} \in \Delta_T \subset \mathbb{R}^T$ is the weighting scheme for the tasks, i.e., the overall loss is calculated as $\mathcal{L} = \boldsymbol{w}^\top \boldsymbol{L} = \sum_{t=1}^T \alpha_t \mathcal{L}_t$. Each member $k \in [m]$ is associated with parameters $\boldsymbol{\theta}_k \in \mathbb{R}^N$ and weighting $\boldsymbol{w} \in \Delta_T$.

**Preliminaries.** Our goal lies in solving an unconstrained vector optimization problem of minimizing $\boldsymbol{L}(\boldsymbol{y}, \hat{\boldsymbol{y}}) = [\mathcal{L}_1(y_1, \hat{y}_1), \ldots, \mathcal{L}_T(y_T, \hat{y}_T)]^\top$, where $\mathcal{L}_i$ corresponds to the objective function for the $i^{\text{th}}$ task, e.g., cross-entropy loss in case of classification. Constructing an optimal solution for all tasks is often unattainable due to competing objectives. Hence, an alternative notion of optimality is used, as described in Definition 1.

**Definition 1** (Pareto Optimality). *A point $\boldsymbol{x}$ dominates a point $\boldsymbol{y}$ if $\mathcal{L}_t(\boldsymbol{x}) \leq \mathcal{L}_t(\boldsymbol{y})$ for all tasks $t \in [T]$ and $\boldsymbol{L}(\boldsymbol{x}) \neq \boldsymbol{L}(\boldsymbol{y})$. Then, a point $\boldsymbol{x}$ is called Pareto optimal if there exists no point $\boldsymbol{y}$ that dominates it. The set of Pareto optimal points forms the Pareto front $\mathcal{P}_{\boldsymbol{L}}$.*

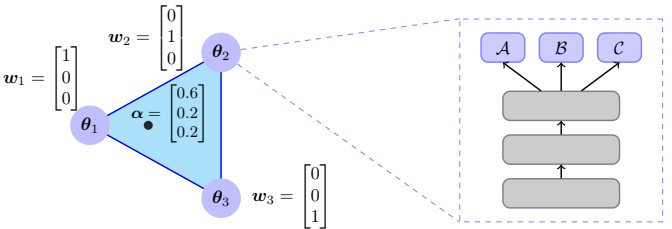

Figure 2: A representation of the encoding in parameter space for $T = 3$ tasks. Each node corresponds to a tuple of parameters and weighting scheme $(\boldsymbol{\theta}_v, \boldsymbol{w}_v) \in \mathbb{R}^N \times \Delta_T$. The blue dashed frame shows the model, e.g., shared-bottom architecture, implemented by the parameters $\boldsymbol{\theta}_v$ of each node. For each training step, we sample $\boldsymbol{\alpha} \in \Delta_T$ and construct the weight combination $\boldsymbol{\theta} = \boldsymbol{\alpha}^\top \boldsymbol{\Theta} = 0.6 \cdot \boldsymbol{\theta}_1 + 0.2 \cdot \boldsymbol{\theta}_2 + 0.2 \cdot \boldsymbol{\theta}_3$.

The vector loss function is scalarized by the vector $\boldsymbol{w} \in [0, 1]^T$ to form the overall objective $\boldsymbol{w}^\top \boldsymbol{L}$. Without loss of generality, we assume that $\boldsymbol{w}$ lies in the $T$-dimensional simplex $\Delta_T$ by imposing the constraint $\|\boldsymbol{w}\| = \sum_{t=1}^T w_t = 1$. This formulation permits to think of the vector of weights as an encoding of task preferences, e.g., for two tasks letting $\boldsymbol{w} = [0.8, 0.2]$ results in attaching more importance to the first task. Overall, the Multi-Task Learning problem can be formulated within the Empirical Risk Minimization (ERM) framework for preference vector $\boldsymbol{w}$ and dataset $\mathcal{D} = \{(\boldsymbol{x}, \boldsymbol{y})\}_{i=1}$ as:

$$\min_{\boldsymbol{\theta}} \quad \mathbb{E}_{(\boldsymbol{x}, \boldsymbol{y}) \sim \mathcal{D}} \left[ \boldsymbol{L} \left( \boldsymbol{y}, \boldsymbol{f} \left( \boldsymbol{x}; \boldsymbol{\theta} \right) \right) \right] \tag{1}$$

Our overall goal is to discover a low-dimensional parameterization in weight space that yields a (continuous) Pareto Front in functional space. This desideratum leads us to the following definition:

**Definition 2** (Pareto Subspace). *Let $T$ be the number of tasks, $\mathcal{X}$ the input space, $\mathcal{Y}$ the multi-task output space, $\mathcal{R} \subset \mathbb{R}^N$ the parameter space, $f : \mathcal{X} \times \mathcal{R} \to \mathcal{Y}$ the function implemented by a neural network, and $\boldsymbol{L} : \mathcal{Y} \times \mathcal{Y} \to \mathbb{R}_{>0}^T$ be the vector loss. Let $\{\boldsymbol{\theta}_t \in \mathcal{R} : t \in [T]\}$ be a collection of network parameters and $\mathcal{S}$ the corresponding convex envelope, i.e., $\mathcal{S} = \left\{ \sum_{t=1}^T \alpha_t \boldsymbol{\theta}_t : \sum_{t=1}^T \alpha_t = 1 \text{ and } \alpha_t \geq 0, \forall t \right\}$. Consider the dataset $\mathcal{D} = (\mathcal{D}_\mathcal{X}, \mathcal{D}_\mathcal{Y})$. Then, the subspace $\mathcal{S}$ is called Pareto if its mapping to functional space via the network architecture $f$ forms a Pareto Front $\mathcal{P} = \boldsymbol{L}(f(\mathcal{D}_\mathcal{X}; \mathcal{S}), \mathcal{D}_\mathcal{Y}) = \{l : l = \boldsymbol{L}(f(\mathcal{D}_\mathcal{X}; \boldsymbol{\theta}), \mathcal{D}_\mathcal{Y}), \quad \forall \boldsymbol{\theta} \in \mathcal{S}\}$.*

## 4    METHOD

We seek to find a collection of $m$ neural network models, of identical architecture, whose linear combination in *weight space* forms a continuous Pareto Front in *objective space*. Model $i$ corresponds to a tuple of network parameters $\boldsymbol{\theta}_i$ and task weighting $\boldsymbol{w}_i$ and implements the function $\boldsymbol{f}(\cdot; \boldsymbol{\theta}_i)$. We impose connectivity among models by modeling the subspace in the convex hull of the ensemble members. Section 4.1 presents the core of the algorithm, and in Section 4.2 we discuss various improvements that address Multi-Task Learning challenges.

### 4.1    PARETO MANIFOLD LEARNING

Let $\boldsymbol{\Theta} = [\boldsymbol{\theta}_1, \boldsymbol{\theta}_2, \ldots, \boldsymbol{\theta}_m]^\top$ be an $m \times N$ matrix storing the parameters of all models, $\boldsymbol{W} = [\boldsymbol{w}_1, \ldots, \boldsymbol{w}_m]^\top$ be a $m \times T$ matrix storing the task weighting of ensemble members. By designing the subspace as a simplex, the objective now becomes:

$$\mathbb{E}_{(\boldsymbol{x}, \boldsymbol{y}) \sim \mathcal{D}} \left[ \mathbb{E}_{\boldsymbol{\alpha} \sim P} \left[ \boldsymbol{\alpha}^\top \boldsymbol{W} \boldsymbol{L} \left( \boldsymbol{y}, \boldsymbol{f} \left( \boldsymbol{x}; \boldsymbol{\alpha} \boldsymbol{\Theta} \right) \right) \right] \right] \tag{2}$$

where P is the sampling distribution placed upon the simplex. In the case where the ensemble members are single-task predictors ($\boldsymbol{w}$ is one-hot) and the number of tasks coincides with the number of ensemble members ($m = T$), the matrix of task weightings $\boldsymbol{W}$ is an identity matrix and Equation 2 simplifies to $\mathbb{E}_{(\boldsymbol{x}, \boldsymbol{y}) \sim \mathcal{D}} \left[ \mathbb{E}_{\boldsymbol{\alpha} \sim P} \left[ \boldsymbol{\alpha}^\top \boldsymbol{L} \left( \boldsymbol{y}, \boldsymbol{f} \left( \boldsymbol{x}; \boldsymbol{\alpha} \boldsymbol{\Theta} \right) \right) \right] \right] = \mathbb{E}_{(\boldsymbol{x}, \boldsymbol{y}) \sim \mathcal{D}} \left[ \mathbb{E}_{\boldsymbol{\alpha} \sim P} \left[ \sum_{t=1}^T \alpha_t \mathcal{L}_t \left( \boldsymbol{y}, \boldsymbol{f} \left( \boldsymbol{x}; \sum_{t=1}^T \alpha_t \boldsymbol{\theta}_t \right) \right) \right] \right]$. By using the same weighting for both

---

**Algorithm 1:** `ParetoManifoldLearning`

---

**Input** : matrix of model parameters $\boldsymbol{\Theta} = \begin{bmatrix} \boldsymbol{\theta}_1 & \boldsymbol{\theta}_2 & \cdots & \boldsymbol{\theta}_T \end{bmatrix}^\top$, vector loss function $\boldsymbol{L}$, train set $\mathcal{D}$,
network $f$, distribution parameters $\boldsymbol{p}$, window $W \in \mathbb{N}$, regularization coefficient $(\lambda > 0)$

1   Initialize each $\boldsymbol{\theta}_v$ independently
2   **for** *batch* $(\boldsymbol{x}, \boldsymbol{y}) \subseteq \mathcal{D}$ **do**
3      $\mathcal{V} \leftarrow \varnothing$
4      **for** $i \in \{1, 2, \ldots, W\}$ **do**
5          sample $\boldsymbol{\alpha}_i \sim \text{Dir}(\boldsymbol{p})$
6          $\mathcal{V} \leftarrow \mathcal{V} \cup \boldsymbol{\alpha}_i$
7          $\boldsymbol{\theta}_i \leftarrow \boldsymbol{\alpha}_i^\top \boldsymbol{\Theta}$        `// construct network in convex hull of ensemble members`
8          $\boldsymbol{L}(\boldsymbol{\alpha}_i) = \begin{bmatrix} \mathcal{L}_1(\boldsymbol{\alpha}_i) & \cdots & \mathcal{L}_T(\boldsymbol{\alpha}_i) \end{bmatrix} \leftarrow \text{criterion}(f(\boldsymbol{x}; \boldsymbol{\theta}_i), \boldsymbol{y})$    `// compute losses`
9      **end**
10     construct multi-forward graphs $\mathcal{G}_t = (\mathcal{V}, \mathcal{E}_t)$ for all tasks $t \in [T]$      `// see Section 4.2`
11     $\mathcal{R} \leftarrow \sum_{t=1}^{T} \log \left( \frac{1}{|\mathcal{E}_t|} \sum_{(\boldsymbol{\alpha}_i, \boldsymbol{\alpha}_j) \in \mathcal{E}_t} \exp \left[ \mathcal{L}_t(\boldsymbol{\alpha}_i) - \mathcal{L}_t(\boldsymbol{\alpha}_j) \right]_+ \right)$    `// multiforward regularization`
12     $\mathcal{L}_{\text{total}} \leftarrow \sum_{i=1}^{W} \boldsymbol{\alpha}_i^\top \boldsymbol{L}(\boldsymbol{\alpha}_i) + \lambda \cdot \mathcal{R}$
13     Backpropagate $\mathcal{L}_{\text{total}}$
14     Gradient *descent* on $\boldsymbol{\Theta}$
15   **end**

---

> [Reviewer qexX]: changed line 11 link to section

158   the losses and the ensemble interpolation, we explicitly associate models and task losses with a one-
159   to-one correspondence, infusing preference towards one task rather than the other and guiding the
160   learning trajectory to a subspace that encodes such tradeoffs.

161   Algorithm 1 presents the full training procedure for this ensemble of neural networks, containing
162   modifications discussed in subsequent sections. Figure 1 showcases the algorithm in a toy example.
163   Concretely, at each training step a random $\boldsymbol{\alpha}$ is sampled and the corresponding convex combination
164   of the networks is constructed. This procedure is shown in Figure 2. The batch is forwarded
165   through the constructed network and the vector loss is scalarized by $\boldsymbol{\alpha}$ as well. The procedure is
166   repeated $W$ times at each batch (see Section 4.2) and a regularization term penalizing non-Pareto
167   stationary points is added (line 11).

> [Reviewer qexX]: addressing weakness 2 → mentioned Fig. 1 in the main text.
>
> [Reviewer qexX]: addressing weakness 2 → mentioned Fig. 2 in the main text.
>
> [Reviewer qexX]: Addressing weakness 4: Changed citation from Figure 4.2 to Section 4.2

168   **Claim 3.** *Let* $\{\boldsymbol{\theta}_t^* \in \mathcal{R} : t \in [T]\}$ *be the optimal ensemble parameters retrieved at the end of*
169   *training by Algorithm 1 and let $\mathcal{S}$ be the their convex hull. Then $\mathcal{S}$ is a Pareto Subspace.*

170   Note that we have chosen a convex hull parameterization of the weight space, but there are other
171   options, such as Bezier curves or other nonlinear paths (Wortsman et al., 2021; Draxler et al., 2018).
172   However, the universal approximation theorem implies no loss of generality for our design choice.
173   In practice, Claim 3 is validated by uniformly sampling the discovered subspace and the definition
174   of a *Pareto Subspace* is relaxed to conform to the nonconvex settings of Deep Learning, i.e., points
175   are called Pareto optimal if the characterization holds in an open neighborhood rather than globally.

> [Reviewer qexX]: Addressing weakness 2 about Pareto optimality and (non-)convexity

## 4.2   REGULARIZATION AND BALANCING

177   **Loss and gradient balancing schemes.** A common challenge in Multi-Task Learning is the case
178   where tasks have different loss scales, e.g., consider datasets with regression and classification tasks
179   such as `UTKFace`. Then, using the same weighting $\boldsymbol{\alpha}$ for both the losses and the weight ensem-
180   bling, as presented in Equation 2, the easiest tasks are favored and the important property of scale
181   invariance is neglected. To prevent this, the loss weighting needs to be adjusted. Hence, we pro-
182   pose simple balancing schemes: one loss and one gradient balancing scheme, whose effect is to
183   warp the space of loss weightings. While gradient balancing schemes are applied on the shared
184   parameters, loss balancing also affects the task-specific decoders, rendering the methodologies can
185   be complementary. To avoid cluttering, balancing schemes are not presented in Algorithm 1.

186   In terms of loss balancing, we use a lightweight scheme of adding a normalization coef-
187   ficient to each loss term which depends on past values. Concretely, let $W \in \mathbb{Z}_+$ be a
188   positive integer and $\mathcal{L}_m(\tau_0)$ be the loss of task $m$ in step $\tau_0$. Then, the regularization coef-
189   ficient is $\overline{\mathcal{L}}(\tau_0; W) = \frac{1}{W} \sum_{\tau=1}^{W} \mathcal{L}_m(\tau_0 + 1 - \tau)$ for $\tau_0 \geq W$ resulting in the overall loss
190   $\mathcal{L}_{total} = \boldsymbol{\alpha}_{\tau_0}^\top \widehat{\boldsymbol{L}} = \sum_{t=1}^{T} \alpha_t \frac{\mathcal{L}_t(\tau_0)}{\overline{\mathcal{L}_m(\tau_0; W)}}$. For gradient balancing. let $\boldsymbol{g}_t$ be the gradient of task

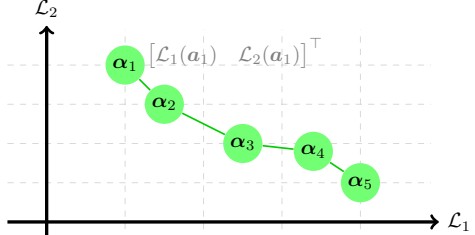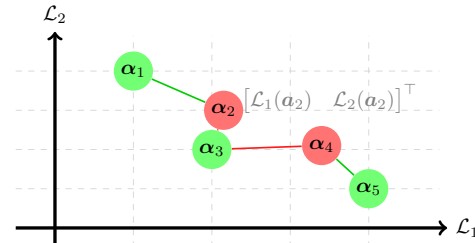

Figure 3: Visual explanation of multiforward regularization, presented in Equation 3. The subfigures depict the loss values for various weightings $\boldsymbol{\alpha}_i = [\alpha_{i,1}, \alpha_{i,2}]$. Optimal lies in the origin. We assume that $\alpha_{1,1} > \cdots > \alpha_{5,1}$. Green color corresponds to Pareto optimality. (Left) all sampled weightings are in the Pareto Front and the regularization term is zero. (Right) The red points are not optimal and, therefore, the regularization term penalizes the violations of the monotonicity constraints for the appropriate task loss: $\boldsymbol{\alpha}_2$ and $\boldsymbol{\alpha}_4$ violate the $\mathcal{L}_1$ and $\mathcal{L}_2$ orderings w.r.t. $\boldsymbol{\alpha}_3$, since $\alpha_{2,1} > \alpha_{3,1} \not\Rightarrow \mathcal{L}_1(\boldsymbol{\alpha}_2) < \mathcal{L}_1(\boldsymbol{\alpha}_3)$ and $\alpha_{4,2} > \alpha_{3,2} \not\Rightarrow \mathcal{L}_2(\boldsymbol{\alpha}_4) < \mathcal{L}_2(\boldsymbol{\alpha}_3)$.

$t \in [T]$ w.r.t. the shared parameters. Previously, the update rule occurred with the overall gradient $\boldsymbol{g}_{total} = \boldsymbol{\alpha}^\top \boldsymbol{G} = \boldsymbol{\alpha}^\top [\boldsymbol{g}_1 \quad \dots \quad \boldsymbol{g}_T]$. We impose a unit $\ell_2$-norm for gradients and perform the update with $\widetilde{\boldsymbol{g}}_{total} = \boldsymbol{\alpha}^\top \widetilde{\boldsymbol{G}} = \boldsymbol{\alpha}^\top [\widetilde{\boldsymbol{g}}_1 \quad \dots \quad \widetilde{\boldsymbol{g}}_T]$ where $\widetilde{\boldsymbol{g}}_t = \frac{\boldsymbol{g}_t}{\|\boldsymbol{g}_t\|_2}$.

**Improving stability by Multi-Forward batch regularization.** Consider two different weightings $\boldsymbol{\alpha}_1$ and $\boldsymbol{\alpha}_2 \in \Delta_{T-1}$. Without loss of generality $[\boldsymbol{\alpha}_1]_0 = \alpha_1 > [\boldsymbol{\alpha}_2]_0 = \alpha_2$. Then, ideally, the interpolated model closer to the ensemble member for task 1 has the lowest loss on that task, i.e., we would want the ordering $\mathcal{L}_1(\boldsymbol{\alpha}_1) < \mathcal{L}_1(\boldsymbol{\alpha}_2)$, and, equivalently for the other tasks. Furthermore, if $\boldsymbol{\alpha} = [1 - \epsilon, \epsilon/T-1, \dots, \epsilon/T-1]$, only one member essentially reaps the benefits of the gradient update and moves the ensemble towards weight configurations more suitable for one task but, perhaps deleterious for the remaining ones. Thus, we propose repeating the forward pass $W$ times for different random weightings $\{\boldsymbol{\alpha}_i\}_{i \in [W]}$, allowing the advancement of all ensemble members concurrently in a coordinated way. By performing multiple forward passes for various weightings, we achieve a lower discrepancy sequence and reduce the variance of such pernicious updates.

We also include a regularization term, which penalizes the wrong orderings and encourages the subspace to have Pareto properties. Let $\mathcal{V}$ be the set of interpolation weighs sampled in the current batch $\mathcal{V} = \{\boldsymbol{\alpha}_w = (\alpha_{w,1}, \alpha_{w,2}, \dots, \alpha_{w,T}) \in \Delta_{T-1}\}_{w \in [W]}$. Then each task defines the *directed* graph $\mathcal{G}_t = (\mathcal{V}, \mathcal{E}_t)$ where $\mathcal{E}_t = \{(\boldsymbol{\alpha}_i, \boldsymbol{\alpha}_j) \in \mathcal{V} \times \mathcal{V} : \alpha_{i,t} < \alpha_{j,t}\}$. The overall loss becomes:

$$\mathcal{L}_{total} = \sum_{i=1}^{W} \boldsymbol{\alpha}_i^\top \boldsymbol{L}(\boldsymbol{\alpha}_i) + \lambda \cdot \sum_{t=1}^{T} \log \left( \frac{1}{|\mathcal{E}_t|} \sum_{(\boldsymbol{\alpha}_i, \boldsymbol{\alpha}_j) \in \mathcal{E}_t} e^{[\mathcal{L}_t(\boldsymbol{\alpha}_i) - \mathcal{L}_t(\boldsymbol{\alpha}_j)]_+} \right) \tag{3}$$

The current formulation of the edge set penalizes heavily the connections from vertices with low values. For this reason, we only keep one outgoing edge per node, defined by the task lexicographic order, resulting in the graph $\mathcal{G}_t^{\text{LEX}} = (\mathcal{V}, \mathcal{E}_t^{\text{LEX}})$ and $|\mathcal{E}_t^{\text{LEX}}| = W - 1, \forall t \in [T]$. Note that the regularization term is convex as the sum of *log-sum-exp* terms. If no violations occur, the regularization term is zero. Figure 3 offers a visual explanation of the regularization term.

**The role of sampling.** Another component of Algorithm 1 is the sampling imposed on the convex hull parameterization. During training, the sampling distribution dictates the loss weighting used and, hence, modulates the degree of task learning. A natural choice is the Dirichlet distribution $\text{Dir}(\boldsymbol{p})$ where $\boldsymbol{p} \in \mathbb{R}_{>0}^T$ are the concentration parameters, since its support is the $T$-dimensional simplex $\Delta_T$. For $\boldsymbol{p} = p\mathbf{1}_T$, the distribution is symmetric; for $p < 1$ the sampling is more concentrated near the ensemble members, for $p > 1$ it is near the centre and for $p = 1$ it corresponds to the uniform distribution. In contrast, for $p_1 \neq p_2$ the distribution is skewed. In our experiments, we use symmetric Dirichlet distributions with $p \geq 1$ to guide the ensemble to representations best suited for Multi-Task Learning.

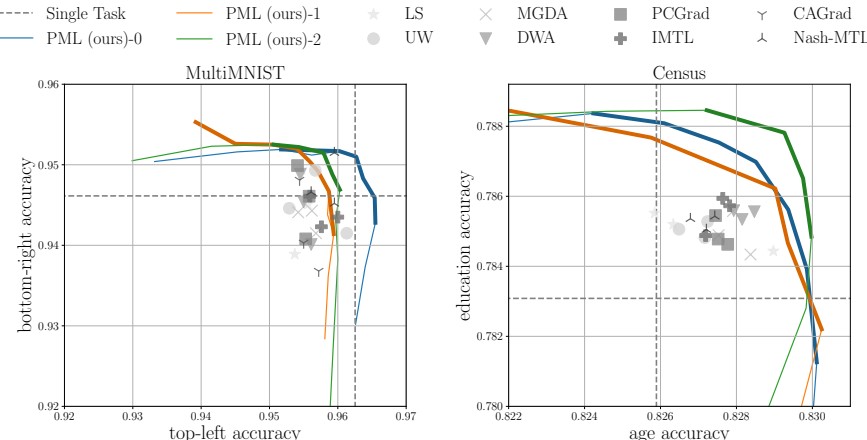

Figure 4: Experimental results on `MultiMNIST` and `Census`. Top right is optimal. Three random seeds per method. Solid lines correspond to our method (PML) and thick lines to the Pareto Front. We have used a different color for each seed of PML. Baselines are shown in shades of gray: scatter plot for MTL baselines and dashed lines for single task. In both datasets, Pareto Manifold Learning discovers subspaces with diverse and Pareto-optimal solutions and outperforms the baselines.

## 5 EXPERIMENTS

We evaluate our method on several datasets, such as `MultiMNIST`, `Census`, `MultiMNIST-3`, `UTKFace` and `CityScapes`, and various architectures, ranging from MultiLayer Perceptrons (MLPs) to Convolutional Neural Networks (CNNs) and Residual Networks (ResNets). Each ensemble member is initialized independently. In all experiments, the learning rate for our method is $m$-fold the learning rate of the baselines to counteract the fact that the backpropagation step scales the gradients by $m^{-1}$ in expectation. The detailed settings used for each dataset and additional experiments are provided in the appendix. Our overarching objective is to construct continuous weight subspaces which map to Pareto Fronts in the functional space. However, our method produces a continuum of results rather than a single point, rendering tabular presentation cumbersome. For this reason, (a) for tables we present the best-of-(sampled)-subspace results, (b) we experiment on numerous two-task datasets where plots convey the results succinctly, (c) present qualitative results on three-task datasets. The source code will be released after the review process.

**Baselines.** We explore various algorithms from the literature: 1. Single-Task Learning (STL), 2. Linear Scalarization (LS) which minimizes the average loss $\frac{1}{T}\sum_{t=1}^{T}\mathcal{L}_t$, 3. Uncertainty Weighting (UW, Cipolla et al. 2018), 4. Multiple-gradient descent algorithm (MGDA, Sener & Koltun 2018), 5. Dynamic Weight Averaging (DWA, Liu et al. 2019), 6. Projecting Conflicting Gradients (PCGrad, Yu et al. 2020), 7. Impartial Multi-Task Learning (IMTL, Liu et al. 2020), 8. Conflict-Averse Gradient Descent (CAGrad, Liu et al. 2021) and 9. Bargaining Multi-Task Learning (Nash-MTL, Navon et al. 2022).

### 5.1 EXPERIMENTS ON DATASETS WITH TWO CLASSIFICATION TASKS

In this section, we focus on datasets with two tasks, both classification. This setting allows for rich visualizations that we use to draw insights on the inner workings of the algorithms.

**MultiMNIST.** We investigate the effectiveness of Pareto Manifold Learning on digit classification using a LeNet model with a shared-bottom architecture. The ensemble consists of two members with single task weightings. To gauge the performance of the models lying in the linear segment between the nodes, we test the performance on the validation set on the ensemble members as well as for 9 models uniformly distributed across the edge, resulting in 11 models in total. We use this evaluation/plotting scheme throughout the experiments. We ablate the effect of multi-forward training on Appendix D; we use a grid search on window $W \in \{2, 3, 4, 5\}$ and strength $\lambda \in \{0, 2, 5, 10\}$ along with the base case of $(W, \lambda) = (1, 0)$ and present in the main text the setting that achieves the

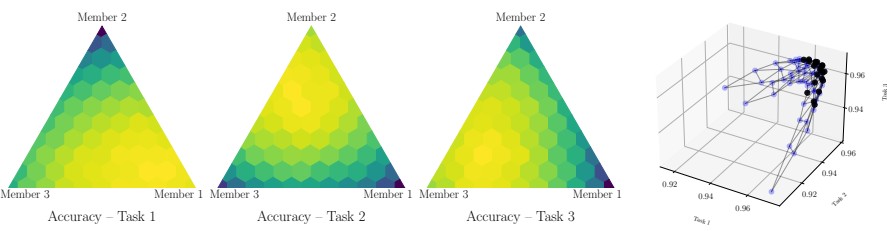

(a) `MultiMNIST-3`: Accuracy Heatmap and Pareto Front for all tasks.

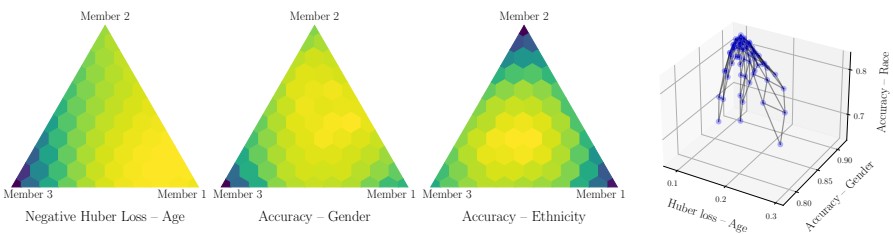

(b) `UTKFace`: Objective Heatmap and Pareto Front for all tasks.

Figure 5: Application of Pareto Manifold Learning on datasets with 3 tasks. Each triangle depicts the performance on a task, using color, as a function of the interpolation weighting, i.e. each hexagon corresponds to a different weighting $\boldsymbol{\alpha} = [\alpha_1, \alpha_2, \alpha_3] \in \Delta_3$. The closer the interpolated member is to a single-task predictor, the higher the performance on the corresponding task. The 3D plot, on the right, show the performance of the model in the multi-objective space.

highest mean (across seeds) HyperVolume score on the validation set. Figure 4 shows the results on `MultiMNIST` using multi-forward regularization with window $W = 4$ and strength $\lambda = 0$. We observe that most baselines are characterized by limited functional diversity; their predefined optimization schemes lead the differently seeded/initialized training runs to final models with similar performance (same markers are clustered in the plots). This lack of functional diversity, as well as inability to consistently outperform the Linear Scalarization baseline, are also noted by Kurin et al. (2022); Xin et al. (2022). In contrast, all Pareto Manifold Learning seeds find subspaces with diverse functional solutions. This statement is quantitatively translated to higher HyperVolume compared to the baselines, shown in Table 4 of the appendix, and can be attributed to the observation that Equation 2 generalizes the Linear Scalarization method.

**Census.** We explore the method on the tabular dataset `Census` (Kohavi, 1996) using a Multi-Layer Perceptron. We focus on the task combination of predicting age and education level, similar to Ma et al. (2020). We perform the same ablation study as before and present the results on Figure 4 for the best setting ($W = 3$ and $\lambda = 10$). In the case of `MultiMNIST`, there exists symmetry between the tasks, both digits are drawn from the same distribution and placed in the pixel grid in a symmetric way, resulting in equal pace learning. However, in the case of `Census`, tasks differ in statistics and, yet, the proposed method recovers a Pareto subspace with diverse solutions.

## 5.2 BEYOND PAIRS OF CLASSIFICATION TASKS: `MultiMNIST-3` AND `UTKFace`

We expand the experimental validation to triplets of tasks, consider regression and more complex architectures, graduating from MLPs and CNNs to ResNets (He et al., 2016). For three tasks, we create a 2D grid of equidistant points spanning the three single-task predictors. If $n$ is the number of interpolated points between two (out of three) members, the grid has $\binom{n+1}{2}$ points. We use $n = 11$, resulting in 66 points. For visual purposes, neighboring points are connected. For three tasks, it would be visually cluttering to present the discovered subspaces with multiple seeds and baselines. Hence, we opt for a more qualitative discussion in this section and present quantitative findings in the appendix.

**MultiMNIST-3.** First, we construct an equivalent of `MultiMNIST` for 3 tasks. Digits are placed on top-left, top-right and bottom-centre. Figure 5a shows the results on `MultiMNIST-3`. As argued previously, `MNIST` variants are characterized by task symmetry and Figure 5a reflects this. For this reason, we do not employ any balancing scheme. The 3D plot in conjunction with the

Table 1: Test performance on *CityScapes*. 3 random seeds per method. For Pareto Manifold Learning, we report the mean (across seeds) best results from the final subspace.

| | Segmentation | | Depth | |
|---|---|---|---|---|
| | mIoU ↑ | Pix Acc ↑ | Abs Err ↓ | Rel Err↓ |
| STL | 71.79 | 92.60 | 0.0135 | 32.786 |
| LS | 70.94 | 92.29 | 0.0192 | 117.658 |
| UW | 70.97 | 92.24 | 0.0188 | 118.168 |
| MGDA | 69.23 | 91.77 | 0.0138 | 51.986 |
| DWA | 70.87 | 92.23 | 0.0190 | 113.565 |
| PCGrad | 71.14 | 92.32 | 0.0185 | 117.797 |
| IMTL | 71.54 | 92.47 | 0.0151 | 65.058 |
| CAGrad | 70.23 | 92.06 | 0.0173 | 100.162 |
| Nash-MTL | 72.07 | 92.61 | 0.0148 | 62.980 |
| PML (ours) | 70.28 | 91.94 | 0.0140 | 52.559 |

simplices reveal that the method has the effect of gradual transfer of learned representation from one member to the other, and offers a succinct visual confirmation of Claim 3.

**UTKFace.** The `UTKFace` dataset (Zhang et al., 2017) has more than 20,000 face images and three tasks: predicting age (modeled as regression using Huber loss - similar to (Ma et al., 2020)), classifying gender and ethnicity. The introduction of a regression task implies that losses have vastly different scales, which dictates the use of balancing schemes, as discussed in Section 4.2. We apply the proposed gradient-balancing scheme and present the results in Figure 5b. For visual unity and to remain in the theme of "higher is better", the *negative* Huber loss is plotted. Despite the increased complexity and the existence of a regression task, the proposed method discovers a *Pareto Subspace*. Additional experiments and qualitative results are provided in Appendix G.

## 5.3 SCENE UNDERSTANDING

We also explore the applicability of Pareto Manifold Learning for `CityScapes` (Cordts et al., 2016), a scene understanding dataset containing high-resolution images of urban street scenes. Our experimental configuration is drawn from Liu et al. (2019); Yu et al. (2020); Liu et al. (2021); Navon et al. (2022) with some modifications. Concretely, we address two tasks: semantic segmentation and depth regression. We use a SegNet architecture (Badrinarayanan et al., 2017) trained for 100 epochs with Adam optimizer (Kingma & Ba, 2015) of initial learning rate $10^{-4}$, which is halved after 75 epochs. The images are resized to $128 \times 256$ pixels. In the initial training steps any sampling $\alpha$ results in a random model, due to initialization, and the algorithm has a warmup period until the ensemble members have acquired meaningful representations. Hence, to reduce computational overhead and help convergence, the concentration parameter of the Dirichlet distribution is set to $p_0 = 5$. We use gradient balancing, window $W = 3$ and $\lambda = 1$. The results are presented in Table 1. In Depth Estimation and out of MTL methods, Pareto Manifold Learning is near-optimal with MGDA narrowly better. However, the performance compared to the other algorithms is superior. In Semantic Segmentation, our method outperforms MGDA, but is worse than other baselines. Overall no multi-task method dominates Pareto Manifold Learning. It is remarkable that, despite our goal of discovering *Pareto subspaces*, the proposed method is on par in performance on Semantic Segmentation with the state-of-the-art algorithms, and better than the vast majority on Depth Estimation.

[Reviewer qexX]: Added short comment addressing weakness 3.

## 6 CONCLUSION

In this paper, we proposed a weight-ensembling method tailored to Multi-Task Learning; multiple single-task predictors are trained in conjunction to produce a subspace formed by their convex hull, and endowed with desirable Pareto properties. We experimentally show on a diverse suite of benchmarks that the the proposed method is successful in discovering *Pareto subspaces* and outperforms some state-of-the-art MTL methods. An interesting future direction is to perform a hierarchical weight ensembling, sharing progressively more of the lower layers, given that the features learned at low depth are similar across tasks. An alternative exploration venue is to connect our method to the challenge of task affinity (Fifty et al., 2021; Standley et al., 2020) via a geometrical lens of the loss landscape.

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

## A  APPENDIX OVERVIEW

As a reference, we provide the following table of contents solely for the appendix.

## B    EXPERIMENTAL DETAILS

**MultiMNIST**    MultiMNIST is a synthetic dataset derived form the samples of MNIST. Since there is no publicly available version, we create our own by the following procedure. For each MultiMNIST image, we sample (with replacement) two MNIST images (of size $28 \times 28$) and place them top-left and bottom-right on a $36 \times 36$ grid. This grid is then resized to $28 \times 28$ pixels. The procedure is repeated 60000 times, 10000 and 10000 times for training, validation and test datasets. We use a LeNet shared-bottom architecture. Specifically, the encoder has two convolutional layers with 10 and 20 channels and kernel size of 5 followed by Maxpool and a ReLU nonlinearity each. The final layer of the encoder is fully connected producing an embedding with 50 features. The decoders are fully connected with two layers, one with 50 features and the output layer has 10. We use Adam optimizer Kingma & Ba (2015) with learning rate $10^{-3}$, no scheduler and the batch size is set to 256. Training lasts 10 epochs.

**Census**    The original version of the Census (Kohavi, 1996) dataset has one task: predicting whether a person's income exceeds $50000. The dataset becomes suitable for Multi-Task Learning by turning one or several features to tasks (Lin et al., 2019). We focus on the task combination of predicting age and education level, similar to Ma et al. (2020). The model has a Multi-Layer Perceptron shared-bottom architecture. The encoder has one layer with 256 neurons, followed by a ReLU nonlinearity, and two decoders with 2 output neurons each (since the tasks are binary classification). Training lasts 10 epochs. We use Adam optimizer learning rate of $10^{-3}$.

**MultiMNIST-3**    The configuration of MultiMNIST is used. Now, the model has three decoders and training lasts 20 epochs.

**UTKFace**    The UTKFace dataset has more than 20,000 face images of dimensions $200 \times 200$ pixels and 3 color channels. The dataset has three tasks: predicting age (modeled as regression using Huber loss - similar to (Ma et al., 2020)), classifying gender and ethnicity (modeled as classification tasks using Cross-Entropy loss). Images are resized to $64 \times 64$ pixels, age is normalized and a 80/20 train/test split is used. We use a shared-bottom architecture; the encoder is a ResNet18 (He et al., 2016) model without the last fully connected layer. The decoders (task-specific layers) consist of one fully-connected layer, where the output dimensions are 1, 2 and 5 for age (modeled as regression), gender (binary classification) and ethnicity (classification with 5 classes). Training lasts 100 epochs, batch size is 256 and we use Adam optimizer with a learning rate of $10^{-3}$. No scheduler is used.

**CityScapes**    Our experimental configuration is very similar to prior work, namely Liu et al. (2019); Yu et al. (2020); Liu et al. (2021); Navon et al. (2022). All images are resized to $128 \times 256$. The tasks used are coarse semantic segmentation and depth regression. The task of semantic segmentation has 7 classes, whereas the original has 19. We use a SegNet architecture (Badrinarayanan et al., 2017) and train the model for 100 epochs with Adam optimizer (Kingma & Ba, 2015) of an initial learning rate $10^{-4}$. We employ a scheduler that halves the learning rate after 75 epochs.

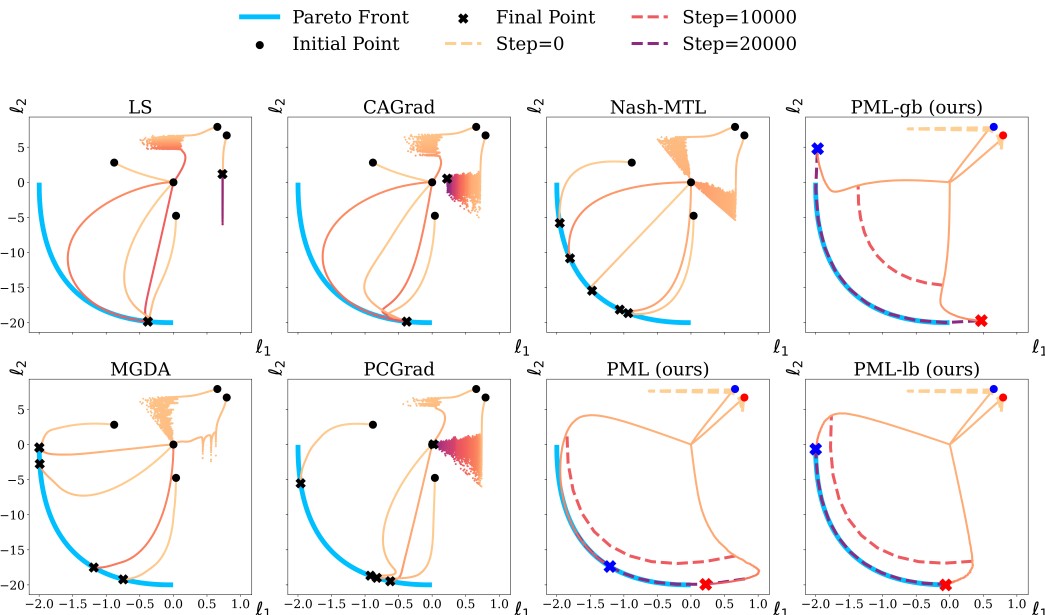

Figure 6: Optimization trajectories in objective space in the case different loss scales. Similar to Figure 1, 5 initializations are shown for baselines and a pair of initializations for Pareto Manifold Learning (PML), in color for clarity. Dashed lines show the evolution of the mapping in loss space for the subspace at the current step. We also show the initial subspace (step= 0). All baselines, except Nash-MTL, and MGDA to a lesser degree, are characterized by trajectories focused on a subset of the Pareto Front, namely minimizing the task with high loss magnitude. The same observation applies to naïvely applying the proposed algorithm PML, because using the same weighting for both the interpolation *and* the losses attaches too much importance on the task with large loss magnitude. However, simple balancing schemes palliate this issue; gradient balancing (PML-gb) discovers a superset of the Pareto Front and loss balancing (PML-lb) discovers the exact Pareto Front.

## C  DETAILS OF THE ILLUSTRATIVE EXAMPLE

The details of the illustrative example are provided in this section. We use the configuration presented by Navon et al. (2022), which was introduced with slight modifications by Liu et al. (2021) and Yu et al. (2020). Specifically, let $\boldsymbol{\theta} = (\theta_1, \theta_2) \in \mathbb{R}^2$ be the parameter vector and $\boldsymbol{L} = (\tilde{\ell}_1, \tilde{\ell}_2)$ be the vector objective defined as follows:

$$\tilde{\ell}_1(\boldsymbol{\theta}) = c_1(\boldsymbol{\theta})f_1(\boldsymbol{\theta}) + c_2(\boldsymbol{\theta})g_1(\boldsymbol{\theta}) \quad \text{and} \quad \tilde{\ell}_2(\boldsymbol{\theta}) = c_1(\boldsymbol{\theta})f_2(\boldsymbol{\theta}) + c_2(\boldsymbol{\theta})g_2(\boldsymbol{\theta})$$

where

$$f_1(\boldsymbol{\theta}) = \log\left(\max\left(\left|0.5\left(-\theta_1 - 7\right) - \tanh\left(-\theta_2\right)\right|, 5e-6\right)\right) + 6,$$
$$f_2(\boldsymbol{\theta}) = \log\left(\max\left(\left|0.5\left(-\theta_1 + 3\right) - \tanh\left(-\theta_2\right) + 2\right|, 5e-6\right)\right) + 6,$$
$$g_1(\boldsymbol{\theta}) = \left(\left(-\theta_1 + 7\right)^2 + 0.1 \cdot \left(-\theta_2 - 8\right)^2\right)/10 - 20,$$
$$g_2(\boldsymbol{\theta}) = \left(\left(-\theta_1 - 7\right)^2 + 0.1 \cdot \left(-\theta_2 - 8\right)^2\right)/10 - 20,$$
$$c_1(\boldsymbol{\theta}) = \max\left(\tanh\left(0.5\theta_2\right), 0\right) \quad \text{and} \quad c_2(\boldsymbol{\theta}) = \max\left(\tanh\left(-0.5\theta_2\right), 0\right)$$

We use the experimental setting outlined by (Navon et al., 2022) with minor modifications, i.e., Adam optimizer with a learning rate of $2e-3$ and training lasts for $50K$ iterations. The overall objectives are $\ell_1 = c \cdot \tilde{\ell}_1$ and $\ell_2 = \tilde{\ell}_2$ where we explore two configurations for the scalar $c$, namely $c \in \{0.1, 1\}$. For $c = 1$, the two tasks have losses at the same scale. For $c = 0.1$, the difference in loss scales makes the problem more challenging and the algo-

rithm used should be characterized by scale invariance in order to find diverse solutions spanning the entirety of the Pareto Front. The initialization points are drawn from the following set $\{(-8.5, 7.5), (0.0, 0.0), (9.0, 9.0), (-7.5, -0.5), (9, -1.0)\}$. In the case of Pareto Manifold Learning with two ensemble members there are $5^2 = 25$ initialization pairs. In the main text we use the initialization pair with the worst initial objective values.

Figure 6 presents the results for the case of different loss scales, i.e., $c = 0.1$. We plot various baselines and three versions of the proposed algorithm, Pareto Manifold Learning or PML in short. We focus on the effect of the balancing schemes, introduced in Section 4.2, resulting in the use of no balancing scheme (denoted as PML), the use of gradient balancing (denoted as PML-gb) and the use of loss balancing (denoted as PML-lb). We dedicate two figures for each version of the algorithm and we present all 25 initialization pairs for completeness. Figure 7 and Figure 8 correspond to no balancing scheme in the case of equal loss scales $c = 1.0$, i.e., they complement Figure 1 of the main text. The subsequent figures focus on the case of unequal loss scales where $c = 0.1$; Figure 9 Figure 10 correspond to no balancing scheme, Figure 11 and Figure 12 correspond to the use of gradient balancing, Figure 13 and Figure 14 correspond to the use of loss balancing. The first figures of each pair show the trajectories for each initialization pair, with markers for initial and final positions. The other figures of each pair dispense of the visual clutter and focus on the subspace discovered in the final step of training, which is plotted with dashed lines along with the analytical Pareto Front in solid light blue. Hence, they provide a succinct overview of whether the method was able or not to discover the (entire) Pareto Front.

For $c = 1.0$, the proposed method is able to retrieve the exact Pareto Front with no balancing scheme for most initialization pairs, as can be seen in Figure 8. In three cases (out of 25), the method fails. In our experiments, we found that allowing longer training times or higher learning rates resolve the remaining cases.s For $c = 0.1$, the problem is more challenging and the vanilla version of the algorithm results in a subset of the analytical Pareto Front. Figure 10 shows that this subset is consistent across initialization pairs, excluding the ones the method fails, and focuses on the task with higher loss magnitude. Applying gradient balancing, shown in Figure 11 and Figure 12, allows the method to retrieve (a superset of) the Pareto Front for all initialization pairs. Similarly, loss balancing, shown in Figure 13 and Figure 14, results in the exact Pareto Front. Hence, the inclusion of balancing schemes endows scale invariance in the proposed algorithm. Balancing schemes are used for the more challenging datasets, such as `CityScapes`.

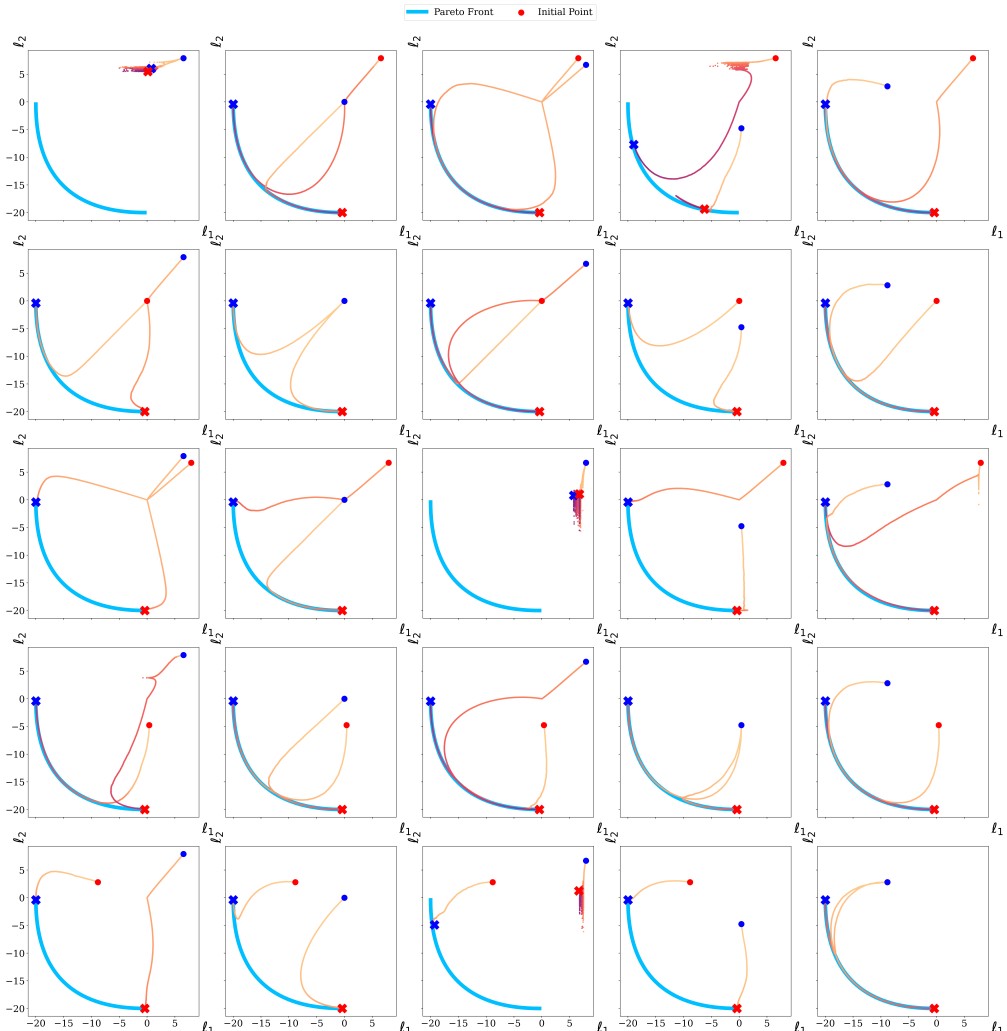

Figure 7: *Illustrative example*. Optimization trajectories in objective space for all initialization pairs in the case of equal loss scales ($c = 1.0$) and application of the proposed method with no balancing scheme. Blue and red markers show each ensemble member's loss value, dots and "X"s correspond to the initial and final step, accordingly. In all but four cases, Pareto Manifold Learning retrieves the entirety of the Pareto Front (can be sen clearly in Figure 8). Allowing longer training times or higher learning rates solves the remaining initialization pairs.

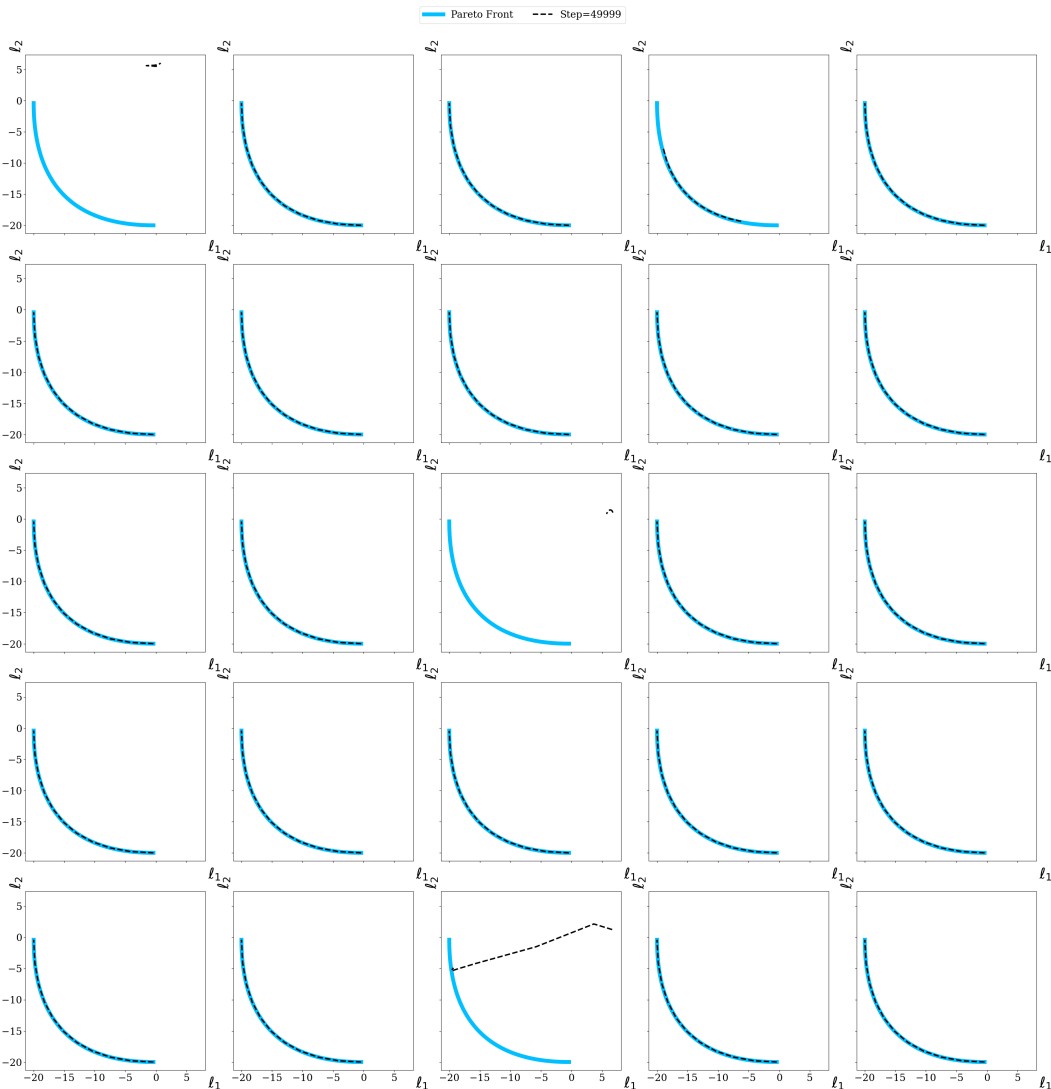

Figure 8: *Illustrative example*. Mapping in objective space of the weight subspace discovered by the proposed method with no balancing scheme, in the case of equal loss scales ($c = 1.0$). The analytic Pareto Front is plotted in light blue. In all but four cases, the dashed line (our method) coincides with the full analytic Pareto Front.

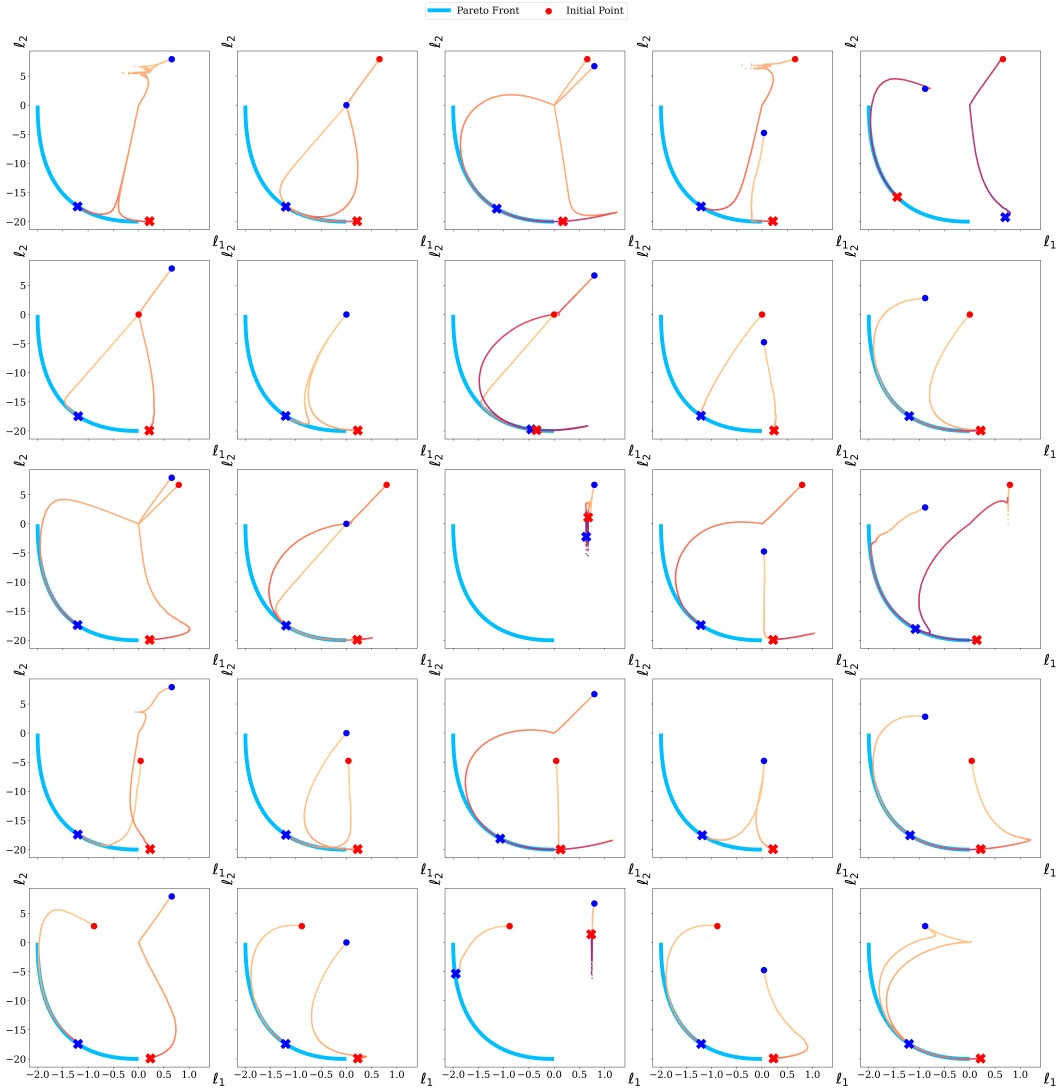

Figure 9: *Illustrative example*. Optimization trajectories in objective space for all initialization pairs in the case of unequal loss scales ($c = 0.1$) and application of the proposed method with no balancing scheme. Blue and red markers show each ensemble member's loss value, dots and "X"s correspond to the initial and final step, accordingly. For the vast majority of initialization pairs, the lack of balancing scheme guides the ensemble to a subset of the Pareto Front, influenced by the task with higher loss magnitude (can be sen clearly in Figure 10).

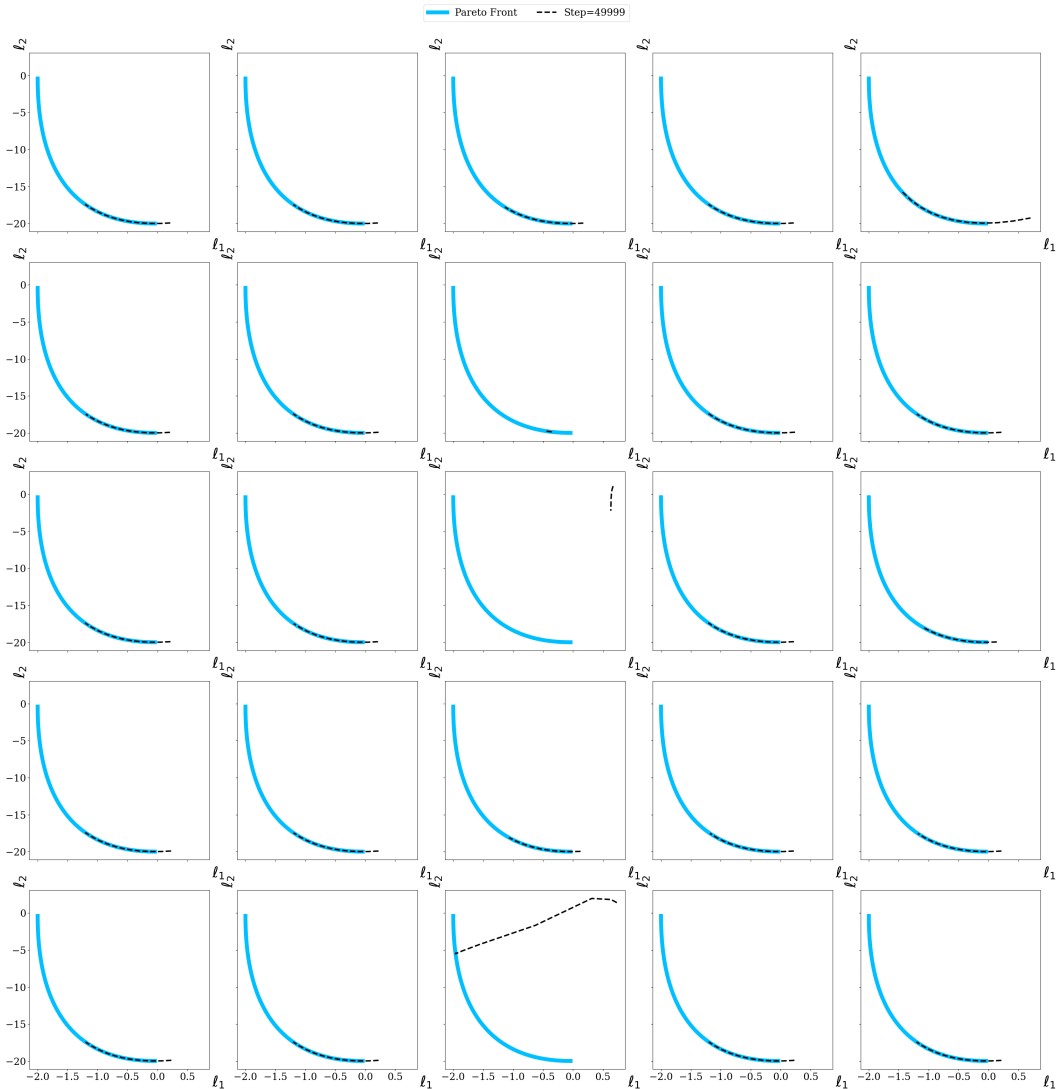

Figure 10: *Illustrative example.* Mapping in objective space of the weight subspace discovered by the proposed method with no balancing scheme, in the case of unequal loss scales ($c = 0.1$). The analytic Pareto Front is plotted in light blue. The lack of balancing scheme renders optimization difficult; the method either completely fails or retrieves a narrow subset of the analytic Pareto Front. Applying balancing schemes resolve these issues.

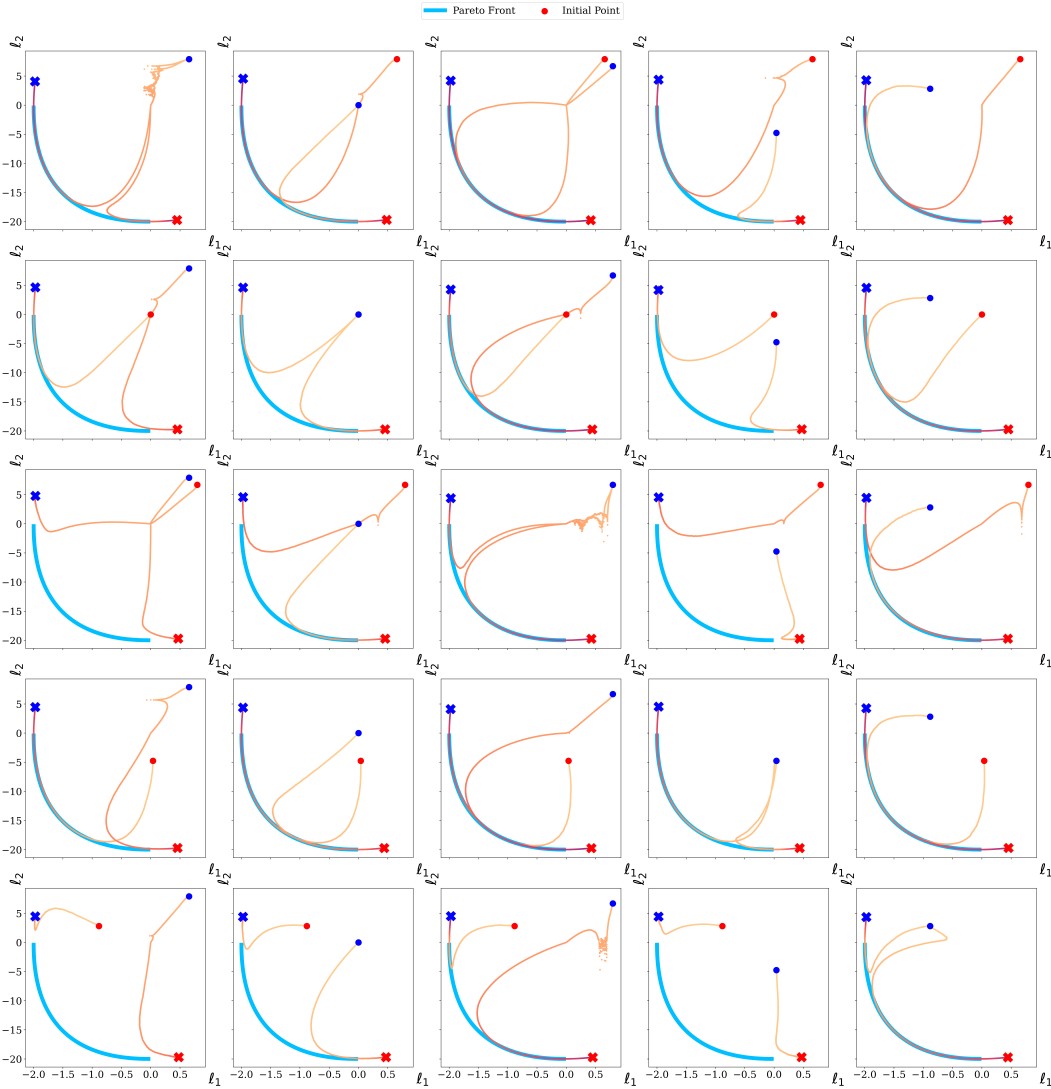

Figure 11: *Illustrative example*. Optimization trajectories in objective space for all initialization pairs in the case of unequal loss scales ($c = 0.1$) and application of the proposed method with gradient balancing scheme. Blue and red markers show each ensemble member's loss value, dots and "X"'s correspond to the initial and final step, accordingly. The proposed method discovers a subspace whose mapping in objective space results in a superset of the Pareto Front. This can be clearly seen in Figure 12.

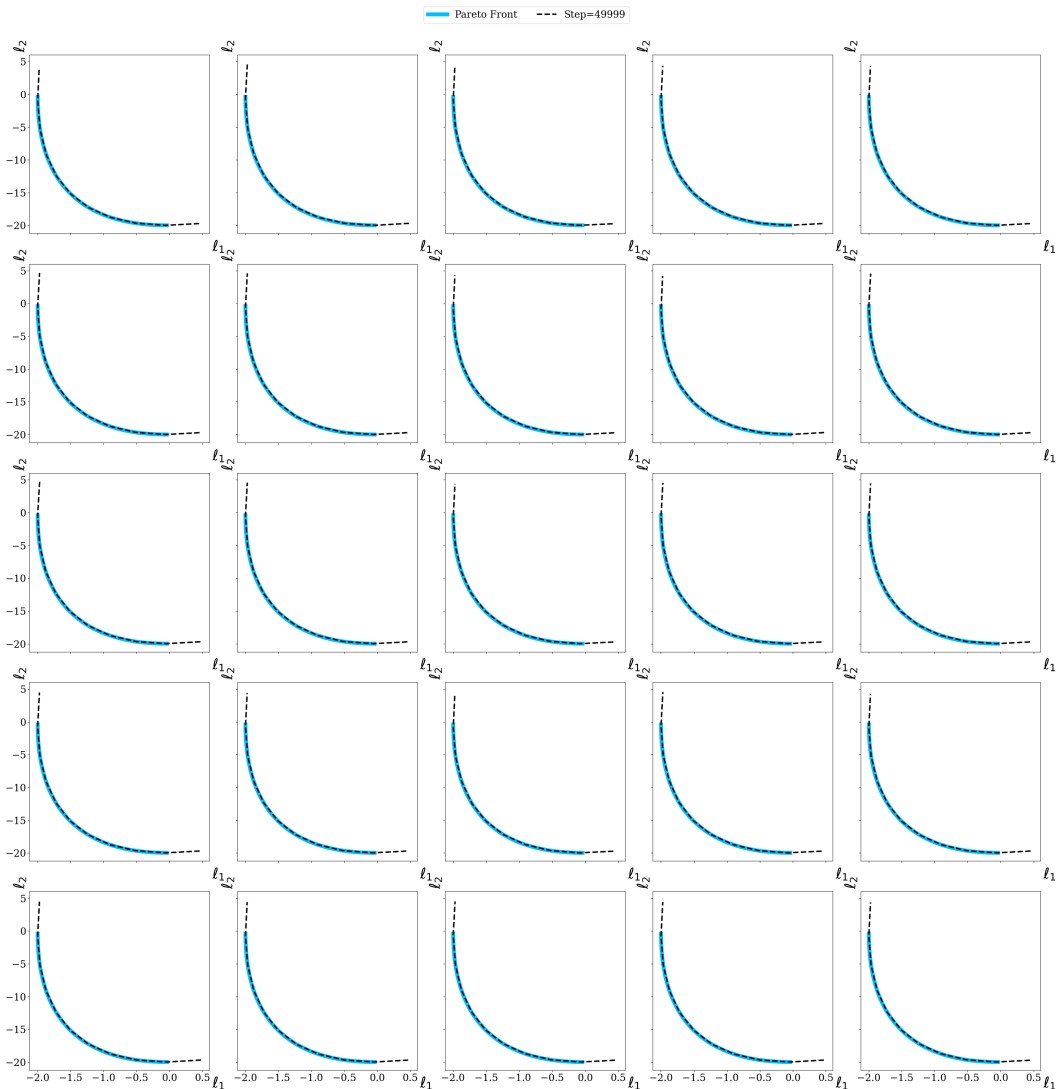

Figure 12: *Illustrative example*. Mapping in objective space of the weight subspace discovered by the proposed method with gradient balancing scheme, in the case of unequal loss scales ($c = 0.1$). The analytic Pareto Front is plotted in light blue. The proposed method consistently finds the same subspace, which is a superset of the analytic Pareto Front.

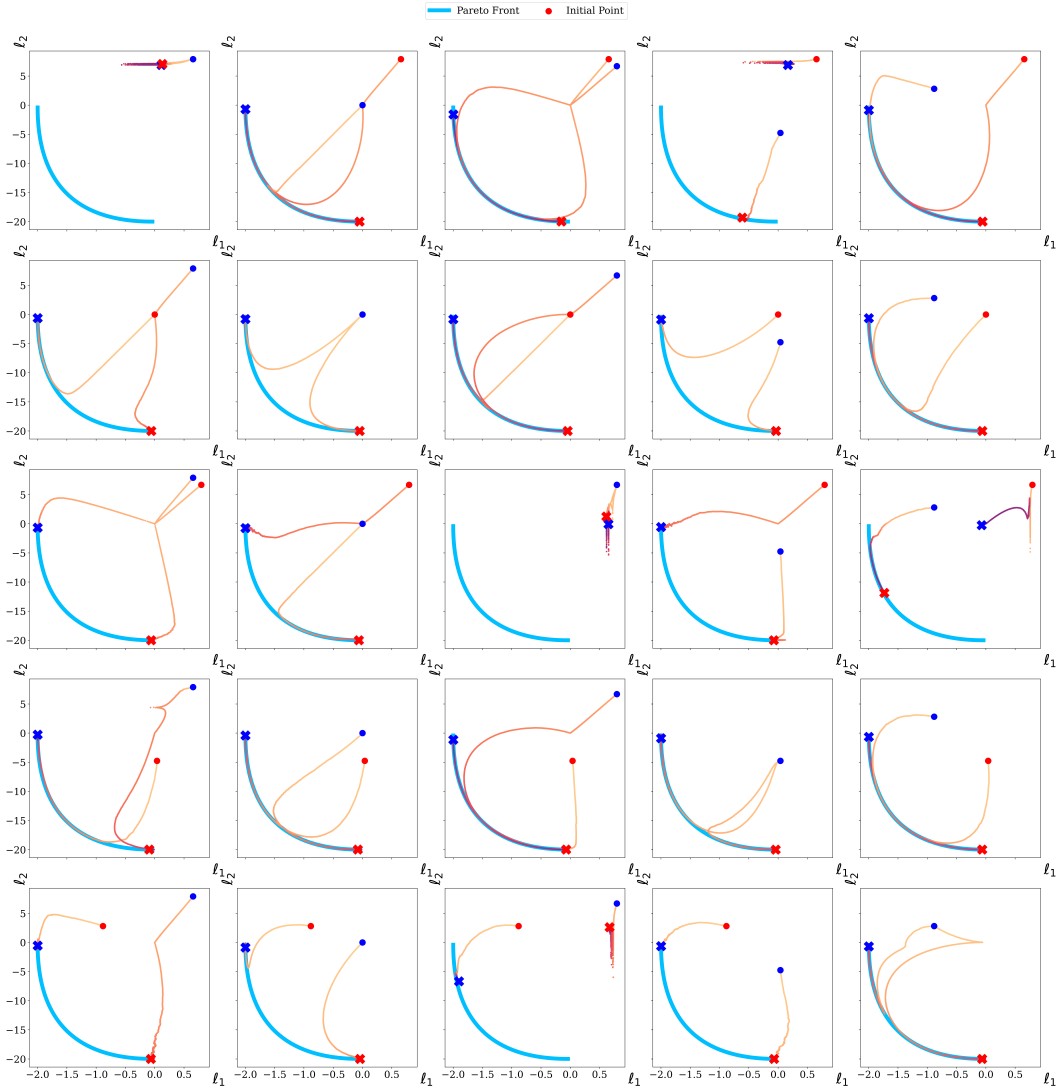

Figure 13: *Illustrative example.* Optimization trajectories in objective space for all initialization pairs in the case of unequal loss scales ($c = 0.1$) and application of the proposed method with loss balancing scheme. Blue and red markers show each ensemble member's loss value, dots and "X"s correspond to the initial and final step, accordingly. For all but five cases, the proposed method discovers a subspace whose mapping in objective space results in the exact Pareto Front. This can be clearly seen in Figure 14.

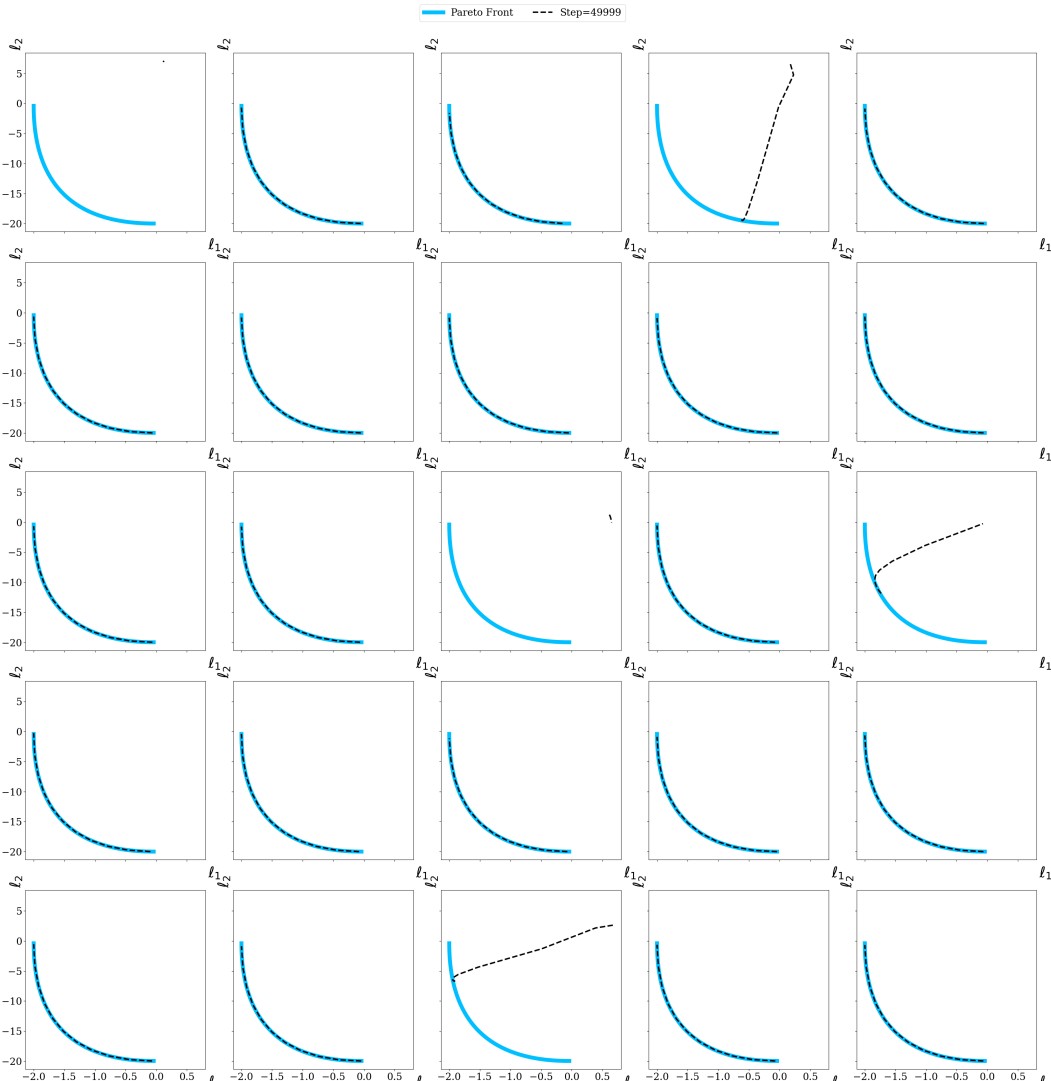

Figure 14: *Illustrative example*. Mapping in objective space of the weight subspace discovered by the proposed method with loss balancing scheme, in the case of unequal loss scales ($c = 0.1$). The analytic Pareto Front is plotted in light blue. Using loss balancing endows scale invariance and the solutions are more functionally diverse, in comparison with no balancing scheme in Figure 10. However, the same initialization pairs continue to be problematic as in the case of equal loss scales (see Figure 8). Allowing for longer training or higher learning rates solves the remaining initialization pairs.

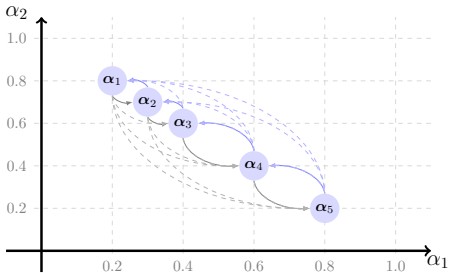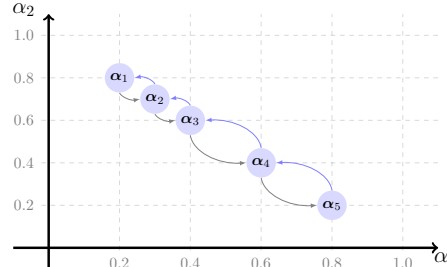

Figure 15: Multi-Forward Graph: case of two tasks. We assume a window of $W = 5$. The nodes lie in the line segment $\alpha_2 + \alpha_1 = 1$, $\alpha_1, \alpha_2 \in [0, 1]$. (Left) Full graph and dashed edges will be removed. (Right) Final graph.

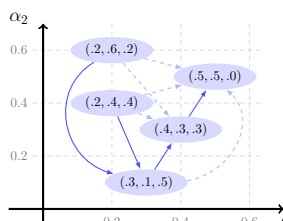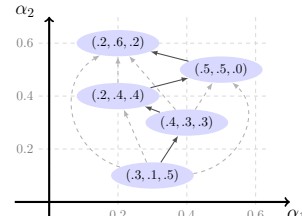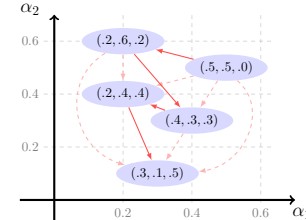

Figure 16: Multi-Forward Graph for three tasks. Left, middle and right present the case of the first, second and third task, respectively. Each node is noted by its weighting, summing up to 1. Edges are drawn if the two nodes obey the total ordering imposed by the task. Dashed edges are omitted from the final graph.

## D    ABLATION ON MULTI-FORWARD REGULARIZATION

Multi-Forward regularization, introduced in Section 4.2, penalizes the ensemble if the interpolated models' losses (sampled within a batch) are not in accordance with the tradeoff imposed by the corresponding interpolation weights. Simply put, the closer we sample to the member corresponding to task 1, the lower the loss should be on task 1. The same applies to the other tasks. Equation 3 in the main text presents the case of two tasks, where the idea of the regularization is outlined in loss space. For completeness, we present the underlying graph construction for the cases of two and three tasks in Figure 15 and Figure 16, respectively. The nodes of the graphs are associated with the sampled weightings and the edges for the graph $\mathcal{G}_t$ of task $t$ are drawn w.r.t. the corresponding partial ordering. If the loss ordering is violated for a given edge, a penalty term is added.

We ablate the effect multi-forward training and the corresponding regularization have on performance. We explore the `MultiMNIST` and `Census` datasets using the same experimental configurations as in the main text. We are interested in two parameters:

- $W$: number of $\boldsymbol{\alpha}$ re-samplings per batch. This parameter is also referred as *window*.
- $\lambda$: the regularization strength as presented in Algorithm 1. For $\lambda = 0$, no regularization is applied but the subspace is still sampled $W$ times and the total loss takes into account all the respective interpolated models.

Figure 17 and Table 2 present the results for `MultiMNIST`. Figure 18 and Table 3 present the results for `Census`. It is important to note that `MultiMNIST` is symmetric, while `Census` is not. As a result, the features learned for each single-task predictor are helpful to one another and the case of $\lambda = 0$, i.e., no regularization and only multi-forward training, is beneficial for `MultiMNIST` but not for `Census`. Intuitively, both digit classification tasks have the same difficulty and posterior distribution, which produces few violations of monotonicity constraints and renders the regularization less applicable. On the other hand, severe regularization such as $\lambda = 10$ can be harmful and hinder training. More details in table and figure captions.

[Reviewer qexX]: expanded commentary on $\lambda = 0$ for `MultiMNIST`.

Table 2: `MultiMNIST`: Ablation on multi-forward training and regularization, presented in [Section 4.2](). Validation performance in terms of HyperVolume (HV) metric. Higher is better, except for standard deviation (std). The visual complement of the table appears in [Figure 17](). For each configuration, we track the Hypervolume across three random seeds and present Mean HV, max HV and standard deviation. We annotate with bold the best per column. In the main text, we report the best result in terms of mean HV, i.e., $W = 4$ and $\lambda = 0$.

|         |                | Seed - 0   | Seed - 1   | Seed - 2   | Mean HV    | Max HV     | std        |
|---------|----------------|------------|------------|------------|------------|------------|------------|
| $W = 2$ | $\lambda = 0$  | 0.9205     | 0.9083     | 0.9100     | 0.9129     | 0.9205     | 0.0054     |
|         | $\lambda = 2$  | 0.9121     | 0.9105     | 0.9037     | 0.9088     | 0.9121     | 0.0036     |
|         | $\lambda = 5$  | 0.9132     | 0.9016     | 0.8979     | 0.9043     | 0.9132     | 0.0065     |
|         | $\lambda = 10$ | 0.8766     | 0.8932     | 0.8470     | 0.8723     | 0.8932     | 0.0191     |
| $W = 3$ | $\lambda = 0$  | 0.9215     | 0.9141     | 0.9111     | 0.9156     | 0.9215     | 0.0044     |
|         | $\lambda = 2$  | 0.9176     | 0.9150     | 0.9122     | 0.9149     | 0.9176     | 0.0022     |
|         | $\lambda = 5$  | 0.9155     | 0.9138     | 0.9140     | 0.9144     | 0.9155     | **0.0008** |
|         | $\lambda = 10$ | 0.9122     | 0.9050     | 0.8962     | 0.9045     | 0.9122     | 0.0066     |
| $W = 4$ | $\lambda = 0$  | **0.9220** | 0.9187     | 0.9143     | **0.9184** | **0.9220** | 0.0032     |
|         | $\lambda = 2$  | 0.9213     | 0.9149     | **0.9157** | 0.9173     | 0.9213     | 0.0028     |
|         | $\lambda = 5$  | 0.9158     | 0.9139     | 0.9132     | 0.9143     | 0.9158     | 0.0011     |
|         | $\lambda = 10$ | 0.9177     | 0.9022     | 0.9102     | 0.9100     | 0.9177     | 0.0063     |
| $W = 5$ | $\lambda = 0$  | 0.9131     | 0.9180     | 0.9156     | 0.9156     | 0.9180     | 0.0020     |
|         | $\lambda = 2$  | 0.9158     | **0.9203** | 0.9146     | 0.9169     | 0.9203     | 0.0024     |
|         | $\lambda = 5$  | 0.9138     | 0.9082     | 0.9140     | 0.9120     | 0.9140     | 0.0027     |
|         | $\lambda = 10$ | 0.9165     | 0.9158     | 0.9121     | 0.9148     | 0.9165     | 0.0019     |

Table 3: `Census`: Ablation on multi-forward training and regularization, presented in [Section 4.2](). Validation performance in terms of HyperVolume (HV) metric. Higher is better, except for standard deviation (std). The visual complement of the table appears in [Figure 18](). For each configuration, we track the Hypervolume across three random seeds and present Mean HV, max HV and standard deviation. We annotate with bold the best per column. In the main text, we report the best result in terms of mean HV, i.e., $W = 2$ and $\lambda = 5$.

|         |                | Seed - 0   | Seed - 1   | Seed - 2   | Mean HV    | Max HV     | std        |
|---------|----------------|------------|------------|------------|------------|------------|------------|
| $W = 2$ | $\lambda = 0$  | 0.6517     | 0.6530     | 0.6532     | 0.6526     | 0.6532     | 0.0006     |
|         | $\lambda = 2$  | 0.6575     | 0.6564     | 0.6560     | 0.6566     | 0.6575     | 0.0006     |
|         | $\lambda = 5$  | **0.6577** | **0.6574** | **0.6590** | **0.6581** | **0.6590** | 0.0007     |
|         | $\lambda = 10$ | 0.6548     | 0.6557     | 0.6554     | 0.6553     | 0.6557     | **0.0004** |
| $W = 3$ | $\lambda = 0$  | 0.6517     | 0.6496     | 0.6501     | 0.6505     | 0.6517     | 0.0009     |
|         | $\lambda = 2$  | 0.6540     | 0.6523     | 0.6544     | 0.6536     | 0.6544     | 0.0009     |
|         | $\lambda = 5$  | 0.6552     | 0.6539     | 0.6536     | 0.6542     | 0.6552     | 0.0007     |
|         | $\lambda = 10$ | 0.6574     | 0.6567     | 0.6566     | 0.6569     | 0.6574     | **0.0004** |
| $W = 4$ | $\lambda = 0$  | 0.6488     | 0.6516     | 0.6504     | 0.6503     | 0.6516     | 0.0011     |
|         | $\lambda = 2$  | 0.6492     | 0.6522     | 0.6504     | 0.6506     | 0.6522     | 0.0012     |
|         | $\lambda = 5$  | 0.6499     | 0.6514     | 0.6525     | 0.6513     | 0.6525     | 0.0011     |
|         | $\lambda = 10$ | 0.6529     | 0.6549     | 0.6558     | 0.6545     | 0.6558     | 0.0012     |
| $W = 5$ | $\lambda = 0$  | 0.6497     | 0.6502     | 0.6484     | 0.6494     | 0.6502     | 0.0008     |
|         | $\lambda = 2$  | 0.6478     | 0.6497     | 0.6495     | 0.6490     | 0.6497     | 0.0009     |
|         | $\lambda = 5$  | 0.6492     | 0.6509     | 0.6489     | 0.6497     | 0.6509     | 0.0009     |
|         | $\lambda = 10$ | 0.6507     | 0.6538     | 0.6508     | 0.6518     | 0.6538     | 0.0014     |

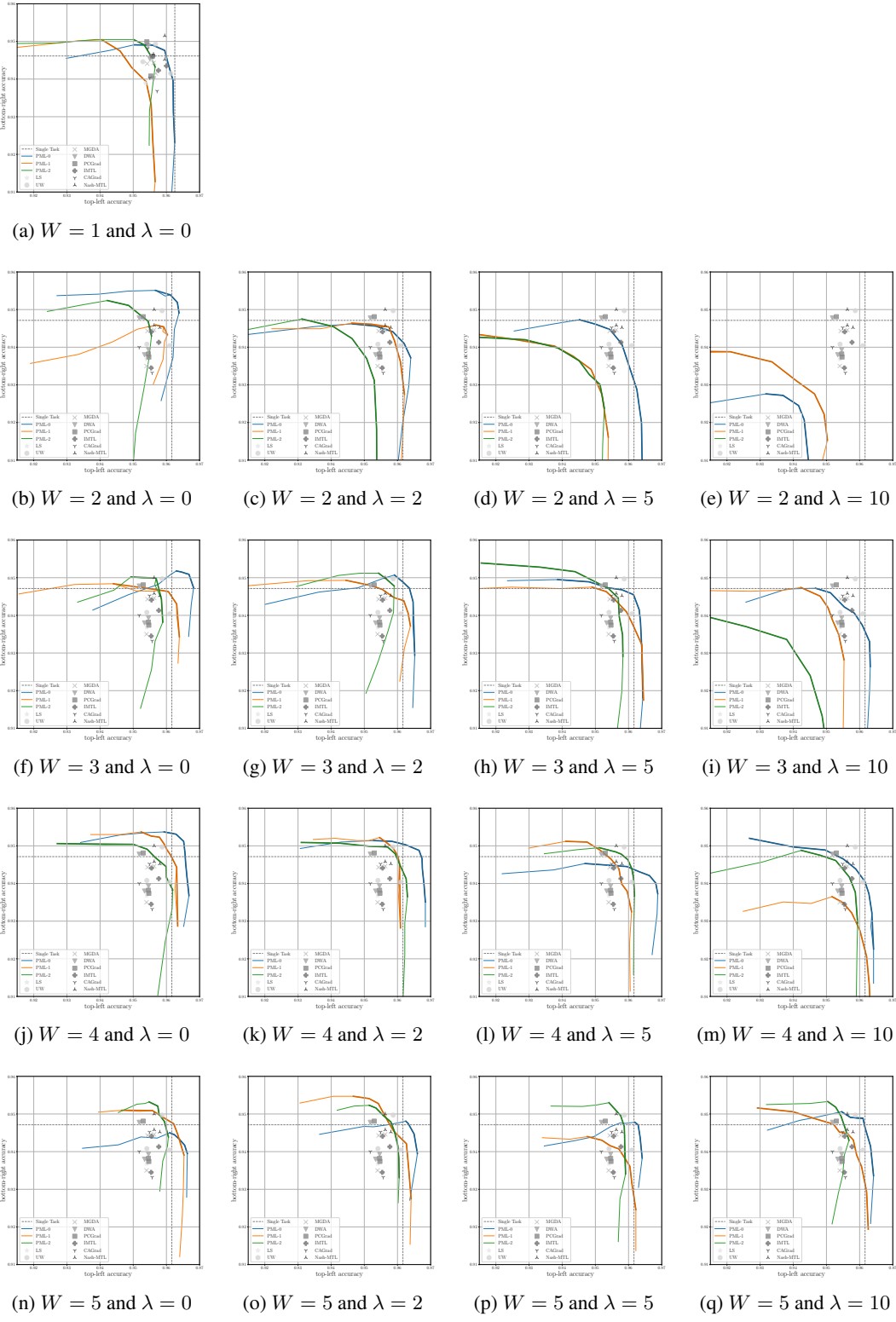

Figure 17: MultiMNIST: Effect of multi-forward on the window $W$ and the regularization coefficient $\lambda$ on the *validation* dataset. The case of no multi-forward ($W = 1$) is presented in the first row. Multi-forward regularization for higher $W$ values is beneficial. Intuitively, attaching serious weight on the regularization $\lambda \in \{5, 10\}$ while sampling few times $W \in \{2, 3\}$ leads to suboptimal performance since the update step focuses on an uninformed regularization term. The accompanying quantitative analysis appears in Table 2.

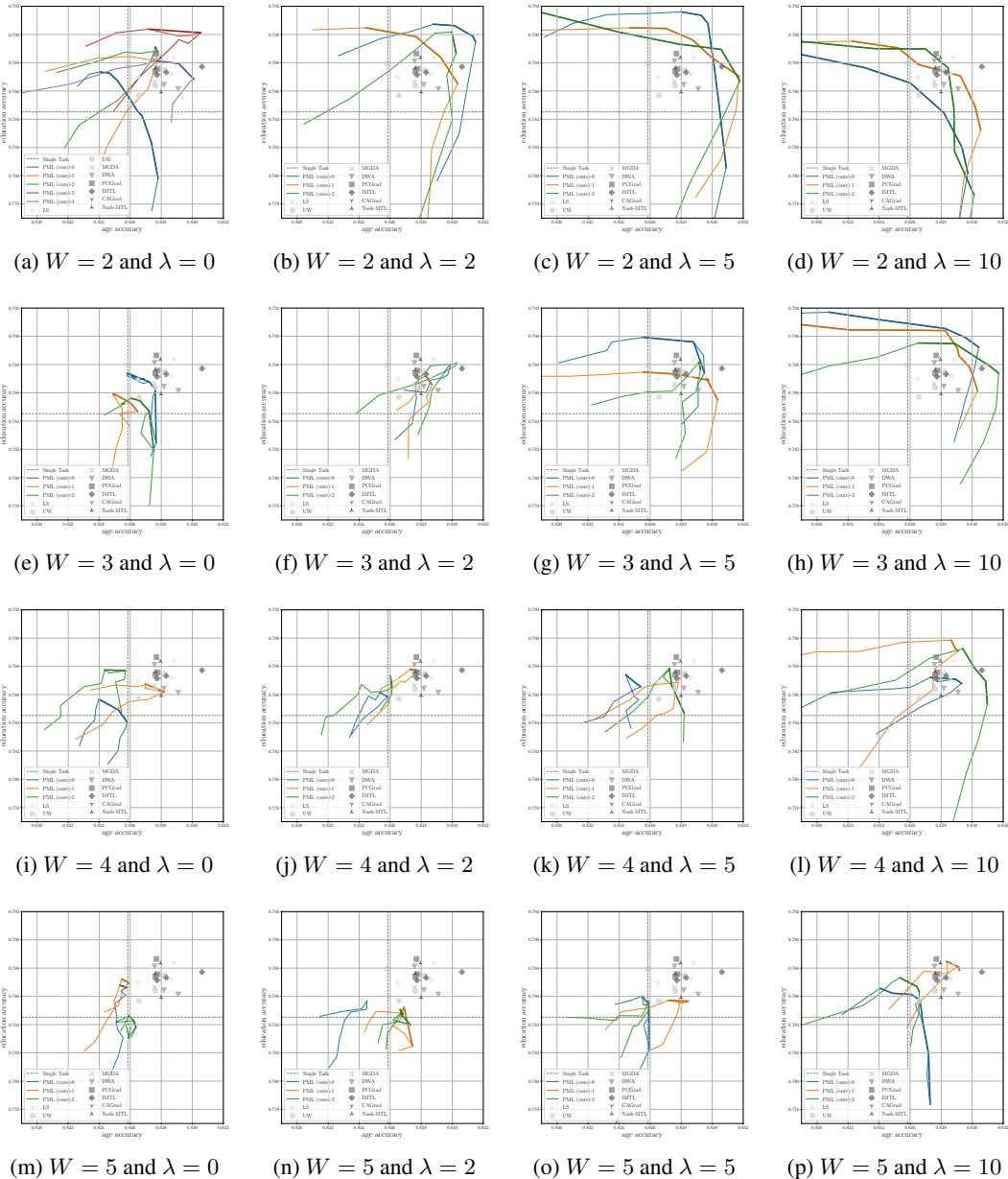

Figure 18: Census: Effect of multiforward on the window $W$ and the regularization coefficient $\lambda$. The axes are shared across plots. Compared to MultiMNIST, applying multiforward on the *asymmetric* Census dataset can improve accuracies and help significantly outperform the baselines. However, widening the window $W$ (e.g., last row for $W = 5$) can be hindering, since larger regularization coefficients are needed. The accompanying quantitative analysis appears in Table 3.

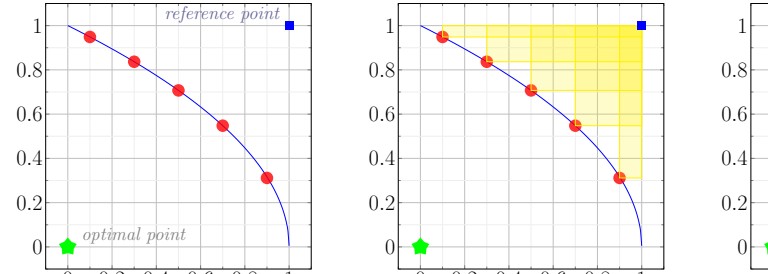

Figure 19: Visual Explanation of Hypervolume. The metric captures the union of axis-aligned rectangles defined by the reference point (star) and the corresponding sample points (red circles). This example showcases loss and the perfect oracle lies in the origin. The point $(1, 1)$ is used for reference. Hence, higher hypervolume implies that the objective space is better explored/covered.

## E    HYPERVOLUME ANALYSIS ON MULTIMNIST AND CENSUS

HyperVolume is a metric widely used in multi-objective optimization that captures the quality of exploration. A visual explanation of the metric is given in Figure 19. Table 4 presents the results of Figure 4 of the main text in a tabular form. We present the best three results per column (higher is better) to succinctly and visually show that all Pareto Manifold Learning seeds outperform the baselines.

Table 4: Tabular complement to Figure 4. Classification accuracy for both tasks and HyperVolume (HV) metric (higher is better). Three random seeds per method. For baselines, we show the mean accuracy and HV (across seeds). For PML, we show the results per seed; HV and max accuracies for the subspace yielded by that seed. We use underlined bold, solely bold and solely underlined font for the best, second best and third best results. We observe that the best results are concentrated in the rows concerning the proposed method (PML). Note that the use of three decimals leads to ties.

|  | MultiMNIST | | | Census | | |
|---|---|---|---|---|---|---|
|  | Task 1 | Task 2 | HV | Task 1 | Task 2 | HV |
| LS | 0.955 | 0.944 | 0.907 | 0.827 | 0.785 | 0.651 |
| UW | 0.957 | 0.945 | 0.913 | 0.827 | 0.785 | 0.650 |
| MGDA | 0.956 | 0.943 | 0.904 | 0.828 | 0.785 | 0.651 |
| DWA | 0.955 | 0.945 | 0.907 | 0.828 | 0.785 | 0.651 |
| PCGrad | 0.955 | 0.946 | 0.908 | 0.828 | 0.785 | 0.650 |
| IMTL | 0.958 | 0.944 | 0.908 | 0.828 | 0.786 | 0.651 |
| Nash-MTL | 0.958 | 0.948 | 0.913 | 0.827 | 0.785 | 0.650 |
| PML - 0 | **_0.968_** | _0.951_ | **_0.92_** | **_0.830_** | **_0.789_** | **_0.655_** |
| PML - 1 | _0.961_ | **_0.953_** | _0.916_ | **_0.830_** | **_0.789_** | **_0.655_** |
| PML - 2 | **0.964** | **_0.953_** | **0.919** | _0.829_ | _0.788_ | _0.653_ |

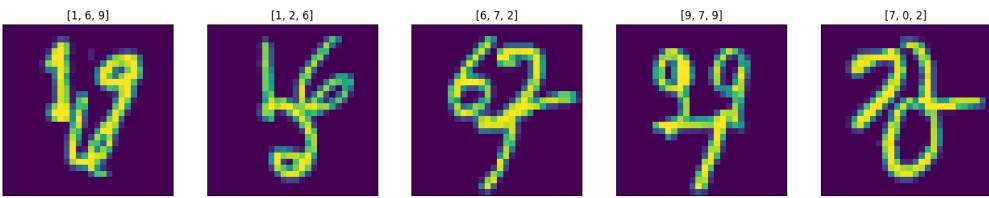

Figure 20: Examples of samples and corresponding labels for the `MultiMNIST-3` dataset.

## F   MULTIMNIST-3 ADDITIONAL RESULTS

This section serves as supplementary to Section 5.2 of the main text. `MultiMNIST-3` is a synthetic dataset generated by `MNIST` samples in a manner similar to the creation of the `MultiMNIST` dataset, which is ubiquitous in the Multi-Task Learning literature. Specifically, each `MultiMNIST-3` sample is created with the following procedure. Three randomly sampled digits of size $28 \times 28$ are placed in the top-left, top-right and bottom middle pixels of a $42 \times 42$ grid. For the pixels where the initial digits overlap, the maximum value is selected. Finally, the image is resized to $28 \times 28$ pixels. Figure 20 shows some examples of the dataset, which consists of three digit classification tasks.

Section 5 compares the performance of baselines and the proposed method while Figure 21 presents visually the performance achieved on the discovered subspace.

Table 5: `MultiMNIST-3`: Mean Accuracy and standard deviation of accuracy (over 3 random seeds). For the proposed method (PML), we report the mean and standard deviation of the best performance from the interpolated models in the sampled subspace. No balancing schemes and regularization are applied. Bold is used for the best performing multi-task method.

[Reviewer qexX]: Table update.

|            | Task 1            | Task 2            | Task 3            |
|------------|-------------------|-------------------|-------------------|
| STL        | $96.97 \pm 0.06$  | $96.10 \pm 0.17$  | $96.40 \pm 0.22$  |
| LS         | $96.26 \pm 0.20$  | $95.48 \pm 0.14$  | $95.87 \pm 0.37$  |
| UW         | $96.48 \pm 0.08$  | $95.42 \pm 0.30$  | $95.77 \pm 0.06$  |
| MGDA       | $96.50 \pm 0.20$  | $94.80 \pm 0.22$  | $95.71 \pm 0.08$  |
| DWA        | $96.42 \pm 0.26$  | $95.26 \pm 0.29$  | $95.75 \pm 0.08$  |
| PCGrad     | $96.45 \pm 0.06$  | $95.39 \pm 0.15$  | $95.88 \pm 0.01$  |
| IMTL       | $96.58 \pm 0.22$  | $95.18 \pm 0.12$  | $96.08 \pm 0.31$  |
| CAGrad     | $96.70 \pm 0.13$  | $95.20 \pm 0.26$  | $95.66 \pm 0.06$  |
| Nash-MTL   | $\mathbf{96.85 \pm 0.08}$ | $95.25 \pm 0.23$  | $96.18 \pm 0.13$  |
| PML (ours) | $\mathbf{96.85 \pm 0.43}$ | $\mathbf{95.72 \pm 0.22}$ | $\mathbf{96.27 \pm 0.32}$ |

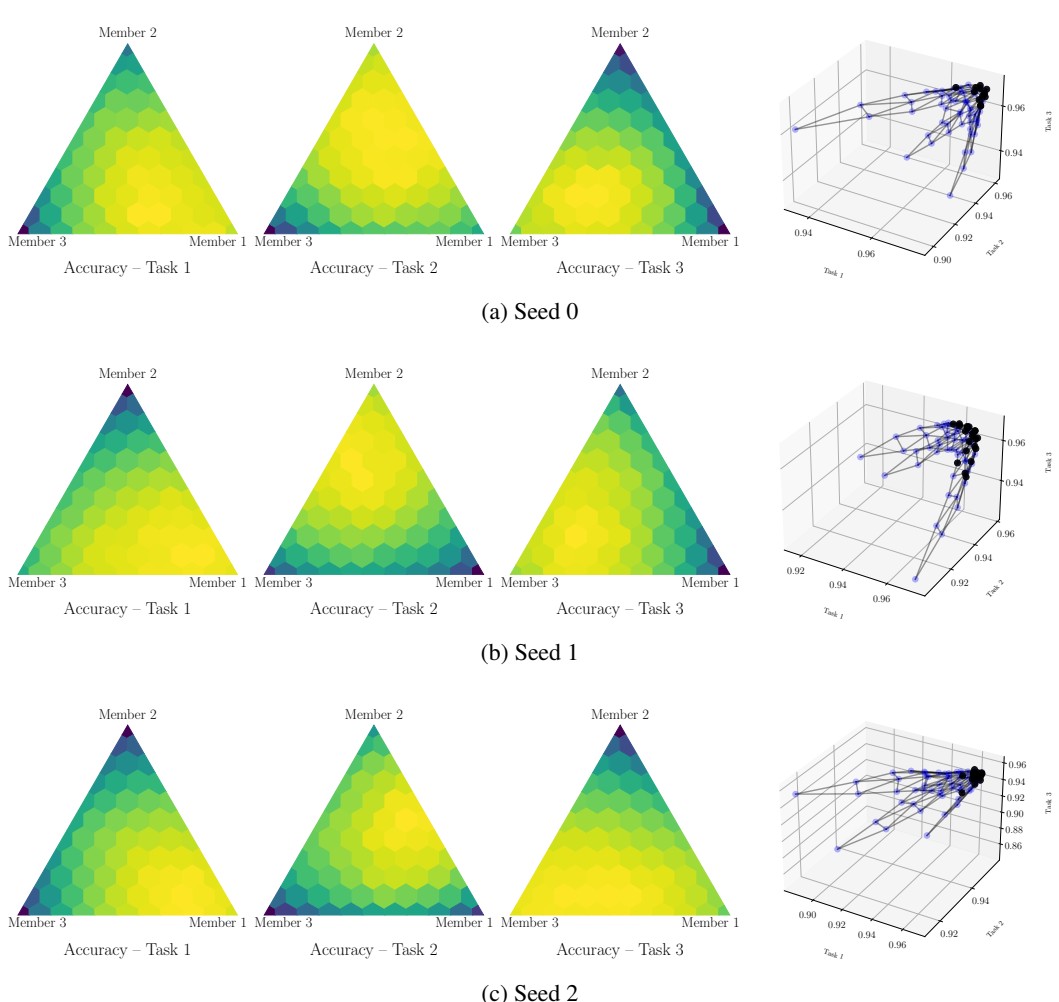

Figure 21: `MultiMNIST-3` results for all three seeds. Each triangle shows the 66 points in the convex hull and color is used for the performance on the associated task. The 3d plot shows the mapping of the subspace to the multi-objective space. No balancing scheme is used.

## G UTKFACE ADDITIONAL RESULTS

This section serves as supplementary to Section 5.2. Section 6 compares the performance of the baselines and the proposed method. We experiment without balancing schemes and with gradient-balancing, and present the results in Figure 22 and Figure 23, respectively. Together with the quantitative results, we observe that for datasets with varying task difficulties, scales, etc. the lack of balancing can be impeding. On the other hand, its inclusion makes the subspace functionally diverse and boosts overall performance. For instance, Huber loss on the task of age prediction is significantly improved.

Table 6: UTKFace: Mean Accuracy and standard deviation of accuracy (over 3 random seeds). For the proposed method (PML), we report the mean and standard deviation of the best performance from the interpolated models in the sampled subspace. No multi-forward training is applied. We present Pareto Manifold Learning with no balancing scheme and with gradient balancing, denoted as *gb*. Bold is used for the best performing multi-task method.

[Reviewer qexX]: Table update.

|  | Age ↓ | Gender ↑ | Ethnicity ↑ |
|---|---|---|---|
| STL | $0.081 \pm 0.005$ | $90.79 \pm 0.55$ | $82.38 \pm 0.40$ |
| LS | $0.086 \pm 0.003$ | $91.66 \pm 0.55$ | $82.78 \pm 0.60$ |
| UW | $0.093 \pm 0.007$ | $91.86 \pm 0.75$ | $83.62 \pm 0.02$ |
| MGDA | $\mathbf{0.075 \pm 0.003}$ | $91.17 \pm 0.59$ | $74.06 \pm 2.66$ |
| DWA | $0.093 \pm 0.008$ | $91.65 \pm 0.46$ | $82.85 \pm 0.20$ |
| PCGrad | $0.101 \pm 0.018$ | $91.85 \pm 0.90$ | $83.57 \pm 0.43$ |
| IMTL | $0.091 \pm 0.004$ | $91.24 \pm 0.34$ | $82.52 \pm 1.15$ |
| CAGrad | $0.083 \pm 0.002$ | $\mathbf{91.93 \pm 0.53}$ | $\mathbf{83.71 \pm 0.33}$ |
| Nash-MTL | $0.095 \pm 0.001$ | $90.40 \pm 0.16$ | $79.59 \pm 0.92$ |
| PML (ours) | $0.096 \pm 0.002$ | $90.97 \pm 0.63$ | $81.78 \pm 0.14$ |
| PML-*gb* (ours) | $0.086 \pm 0.003$ | $91.61 \pm 0.52$ | $81.77 \pm 0.86$ |

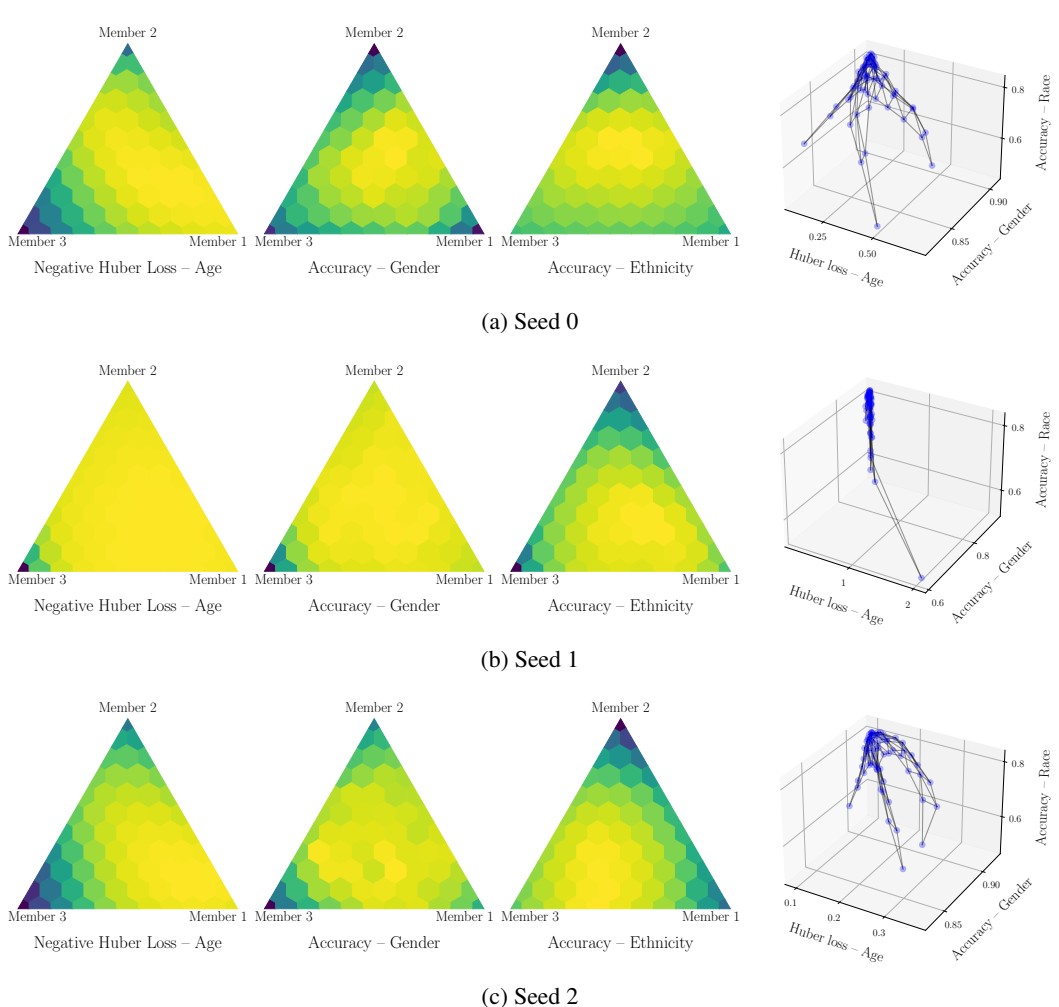

(a) Seed 0

(b) Seed 1

(c) Seed 2

Figure 22: UTKFace results with Linear Scalarization for all three seeds. Each triangle shows the 66 points in the convex hull and color is used for the performance on the associated task. The 3d plot shows the mapping of the subspace to the multi-objective space. Applying no balancing scheme for datasets with different loss scales, e.g., regression and classification tasks, may lead to limited functional diversity, such as for seed 1.

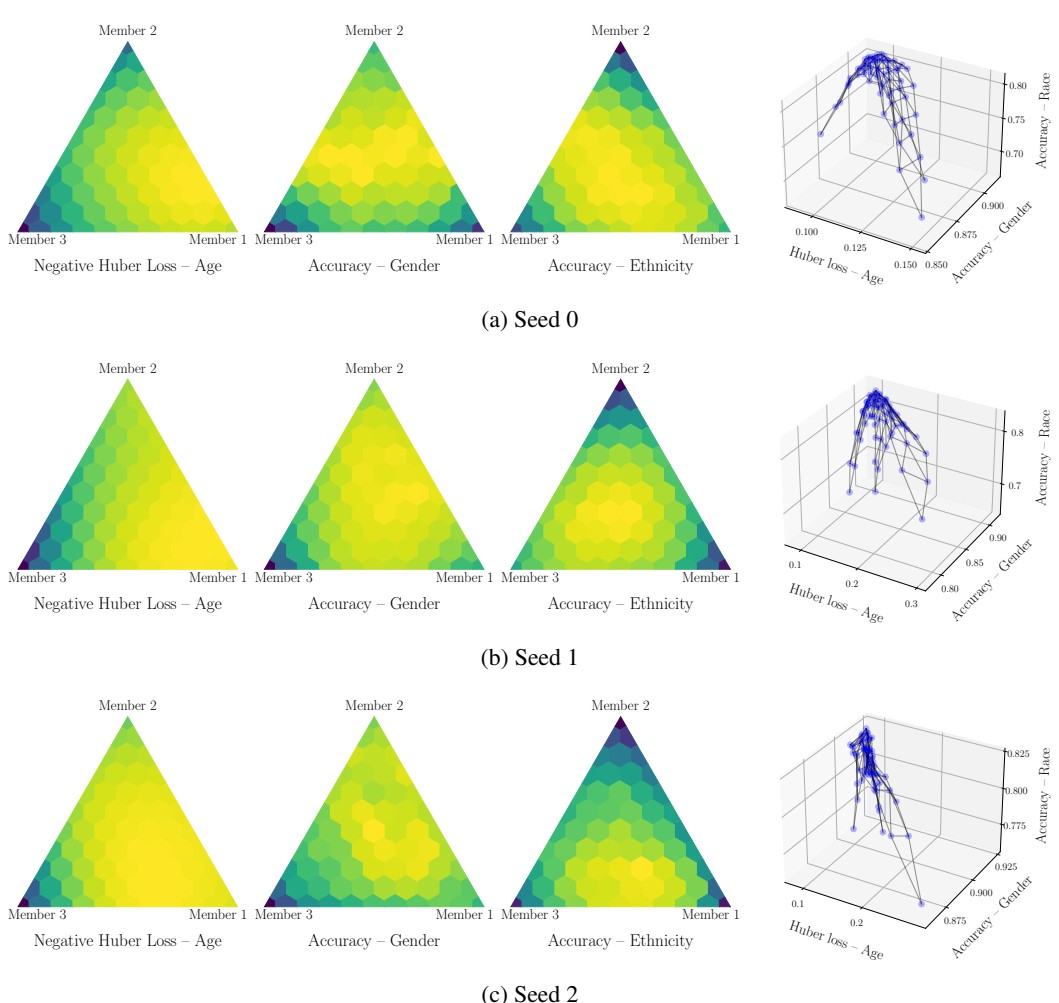

(a) Seed 0

(b) Seed 1

(c) Seed 2

Figure 23: `UTKFace` results with Gradient-Balancing Scheme for all three seeds. Each triangle shows the 66 points in the convex hull and color is used for the performance on the associated task. The 3d plot shows the mapping of the subspace to the multi-objective space. For datasets with tasks of varying loss scales, applying gradient balancing improves functional diversity and performance, as shown in Section 6.

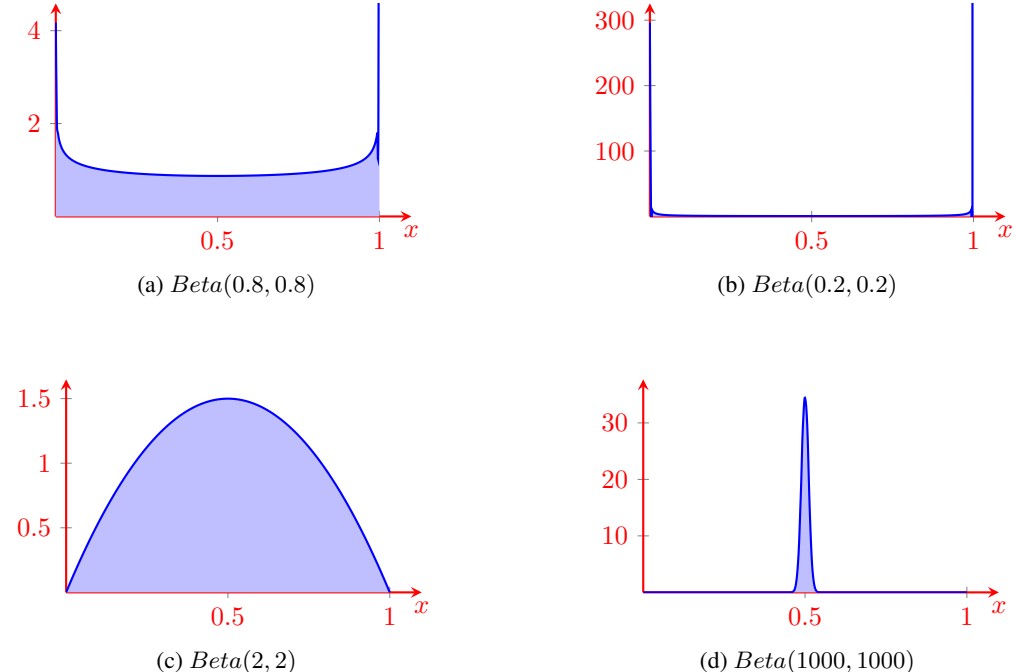

Figure 24: Dirichlet distribution in the case of two tasks. Top row: $p < 1$ and the distribution is more concentrated towards the ensemble members. Bottom row: $p > 1$ and the distribution focuses more on the midpoint which corresponds to all tasks having the same weight. Right column: extreme choices $p \to 0$ or $p \to \infty$. Left column: milder choices.

## H  DETAILS ON SAMPLING

[Reviewer SqFR]: Added appendix regarding sampling.

This appendix expands on Section 4.2 and, specifically, presents in greater detail the intuition behind the sampling distribution's parameters. Let $\boldsymbol{p} \in \mathbb{R}_+^T$ be the parameters of the Dirichlet distribution. Assuming no prior knowledge on the tasks, e.g., task difficulties or affinities, a symmetric distribution is used by setting $\boldsymbol{p} = p\mathbf{1}_T$. This design choice results in three cases:

- $p = 1$: the distribution is uniform on the simplex. Intuitively this means that all tasks are equally important and we care about the diversity of solutions for all tradeoffs (reflected in the linear scalarization weights)

- $p \in (0, 1)$: the distribution is more concentrated towards the ensemble members, as in the top row of Figure 24. Assume an extreme case of two tasks and p0. Then the distribution degenerates to a Bernoulli distribution. Effectively, at each iteration one of the ensemble members is selected and its weights are updated, which will result in two separate and independent single-task predictors with no common representation infused about the other task. Then, linearly interpolating in weight space will result in models with random predictions for both tasks, since the training procedure has not focused in retrieving a Pareto Subspace.
  For milder cases (e.g. $p = 0.7$), we observed that the models in the middle of the linear interpolation suffered in performance which can be attributed to the fact that the sampling focused more on single-task rather than multi-task representations and performance.

- $p > 1$. Then the distribution is more concentrated towards the midpoint of the simplex, as in the bottom row of Figure 24. Assume an extreme case of two tasks and $p \to \infty$. Then, the distribution becomes deterministic and outputs equal weights for all tasks. The randomly and independently initialized ensemble members will collapse to each other, resulting in duplicate ensemble members. Similarly, for very large values (e.g. $p = 100$), the functional diversity of the ensemble will suffer since the weights produced by the distribution will be almost equal for all tasks, resulting in a milder version of the aforementioned phenomenon. In contrast, we found that small values such as $p = 2$ or $p = 3$ can help convergence since

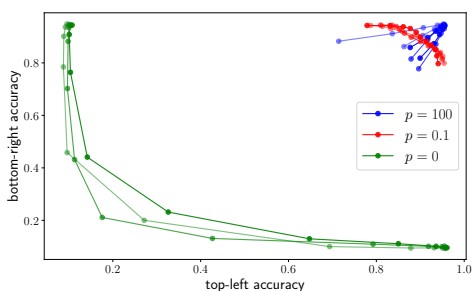
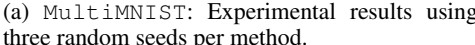

(a) `MultiMNIST`: Experimental results using three random seeds per method.

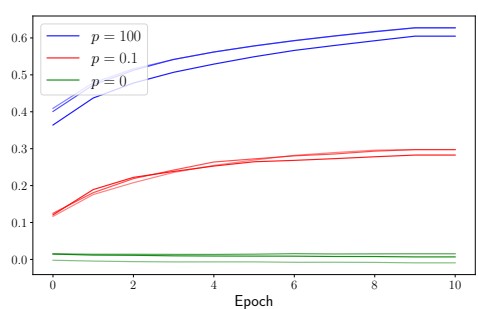

(b) `MultiMNIST`: Cosine similarities of ensemble members.

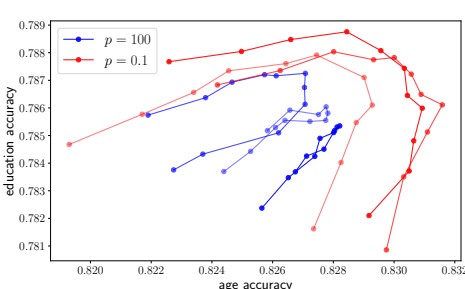

(c) `Census`: Experimental results using three random seeds per method.

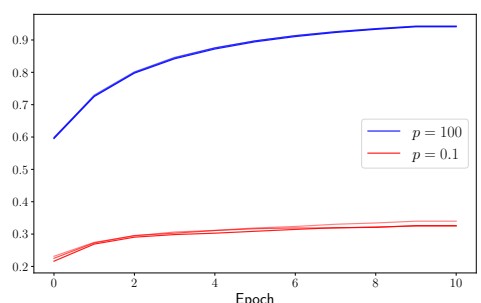

(d) `Census`: Cosine similarities of ensemble members.

Figure 25: Experimental results on `MultiMNIST` and `Census` varying the concentration parameters $p = p\mathbf{1}_T$ of the sampling distribution. Three seeds depicted in shades of the same colors for the various $p$.

they put more emphasis towards common representation (compared to $p = 1$), but may limit functional diversity.

Figure 25 presents experimental results on `MultiMNIST` and `Census` for various concentration parameters $p \in \{0, 0.1, 100\}$ of the Dirichlet distribution. Let $\boldsymbol{\theta}_1$ and $\boldsymbol{\theta}_2$ be the parameters of the ensemble members. For $p = 0$, the ensemble consists of two single-task predictors with no multitask learning representational knowledge, since their interpolation meets a low accuracy/high loss barrier. We omit the case of $p = 0$ for `Census` for visual clarity. This lack of common representation is evident in the cosine similarities as well, where for $p = 0 \cos(\boldsymbol{\theta}_1, \boldsymbol{\theta}_2) \approx 0$. On the other hand, for $p = 0.1$, common representations are infused into the ensemble and the experimental results show that the test performance is characterized by diversity. However, this comes at the expense of the interpolated models at the middle of the line segment, where the performance is suboptimal compared to $p = 100$ for `MultiMNIST`. This behavior is also illustrated in the cosine similarities, where for $p = 100$ the ensemble weights $\boldsymbol{\alpha}$ are in an $\epsilon$-ball around the midpoint causing the independently initialized models to progressively collapse. For `Census`, we also observe that this collapsing leads to very high cosine similarity $\cos(\boldsymbol{\theta}_1, \boldsymbol{\theta}_2) > 0.9$ and the ensemble is suboptimal compared to $p = 0.1$.

## I CONNECTION BETWEEN PARETO OPTIMALITY AND MULTIPLE VALLEY INTERSECTIONS

In this section, we investigate the connection between the intersection of multiple loss landscapes, pareto optimality and the effect of the proposed algorithm Pareto Manifold Learning. We use the illustrative example, presented in Figure 1. Let $\Theta$ be the parameter space of the model and $\mathcal{L}_t : \Theta \to \mathbb{R}, t \in \{1, 2\}$, be the losses of the problem. For $\alpha \in [0, 1]$ and $\theta \in \Theta$, the overall objective is $\mathcal{L}(\theta, \alpha) = \alpha\mathcal{L}_1(\theta) + (1 - \alpha)\mathcal{L}_2(\theta)$.

Figure 26 and the accompanying Figure 27 present the overall loss objective as $\alpha$ varies from 0 to 1. For the extreme values of the range, the loss landscape is inherently single-task. The subspace discovered by the method is depicted in blue, while a black 'x' is used for the corresponding interpolated model, i.e., it corresponds to $\mathcal{L}(\alpha\theta_1 + (1 - \alpha)\theta_2, \alpha)$. Figure 28 presents the overall losses on the subspace by fixing as a function of one of the parameters. In other words, the proposed method tracks the optimum in parameter space as the overall objective evolves and the various loss landscapes are weighted accordingly. While an acceptable multi-task solution lies in the intersection of low loss landscapes, Pareto Manifold Learning focuses on the aforementioned dynamic scenario of loss weighting.

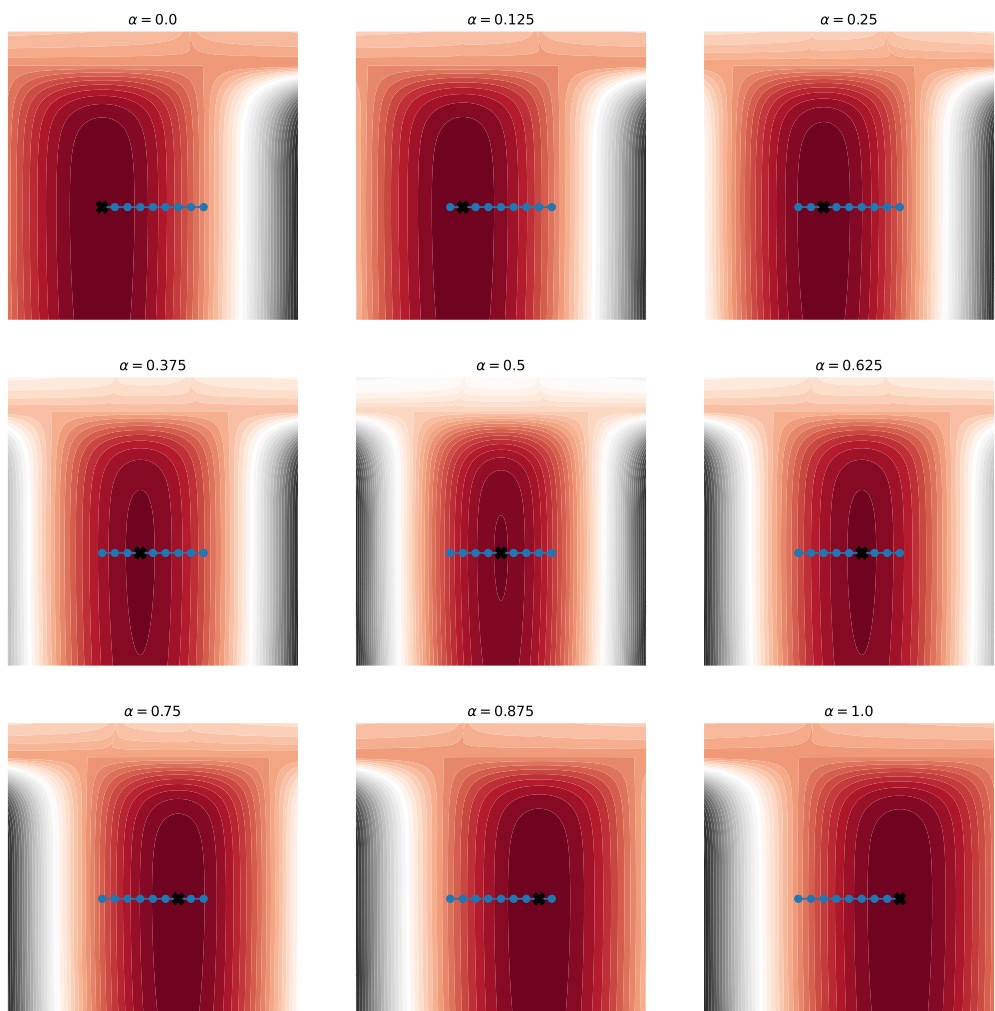

Figure 26: *Illustrative example*: (Overall) loss surface as a function of the model's weights. The overall objective is $\mathcal{L}(\theta, \alpha) = \alpha\mathcal{L}_1(\theta) + (1 - \alpha)\mathcal{L}_2(\theta)$ and is shown for various values of $\alpha$. The Pareto subspace discovered by the proposed method is depicted in blue. 'X' shows the solution of the method for the corresponding $\alpha$.

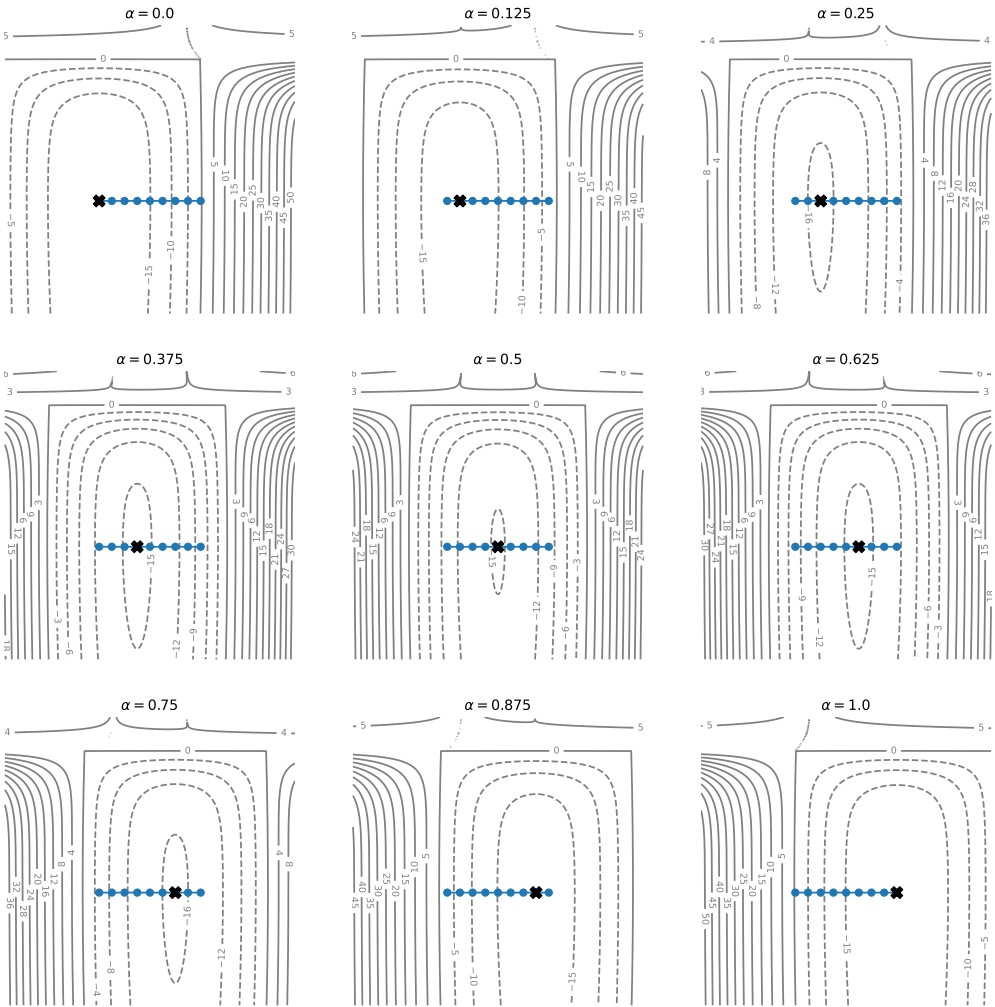

Figure 27: *Illustrative example*: Alternate view of Figure 26. Refer to the text for details.

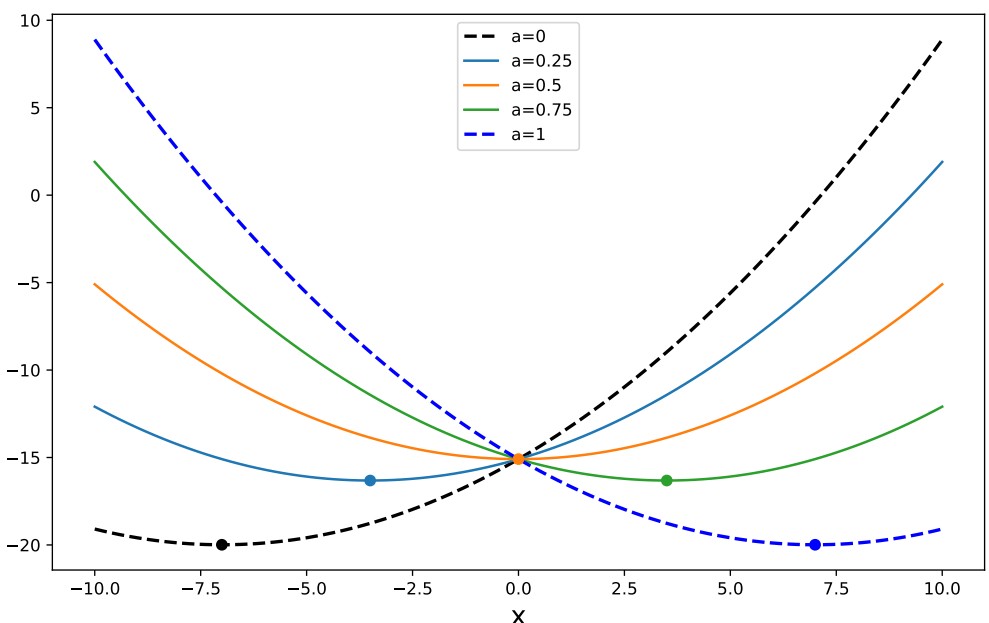

Figure 28: *Illustrative example*: Overall loss for various weightings $\alpha \in [0, 1]$ as a function of one of the parameters, denoted as $x$. Points corresponds to the loss achieved by the parameter vector $\boldsymbol{\theta}(\alpha) = \alpha\boldsymbol{\theta}_1 + (1 - \alpha)\boldsymbol{\theta}_2$. The subspace discovered by the model spans the range $[-7, 7]$.

## J  ADDITIONAL RELATED WORK

In this appendix, we further expand on prior work. Linear mode connectivity, as in (Wortsman et al., 2021), encourages flatness and, therefore, is linked with methods explicitly enforcing flat minima (Chaudhari et al., 2017; Foret et al., 2021; Dinh et al., 2017; Jiang* et al., 2020). These approaches are applicable when designing a single objective, e.g. average of losses in Multi-Task Learning, but do not allow for the infusion of Pareto properties and the inclusion of tradeoffs. Izmailov et al. (2018) produce flat minima by averaging multiple weight vectors discovered during the optimization trajectory, so that the final model lies in the middle of the low-loss basin. Wortsman et al. (2022) perform weight ensembling with fine-tuned models produced via different hyperparameter configurations. Apart from the recent weight ensembling works, output ensembling has been one of the staples of machine learning literature. Lakshminarayanan et al. (2017) utilize deep ensembles for uncertainty prediction but inference scales linearly with the number of ensemble members. Wen et al. (2020) improve on the computational complexity of output ensembles by sharing the bulk of the parameters among members and differentiating them via rank-1 matrices, while Havasi et al. (2021) employ a multi-input multi-output network by accommodating independent subnetworks for each ensemble and allowing a single-forward pass ensemble prediction. However, this results in subnetworks with incompatible architecture which does not allow for a continuous approximation of the Pareto Front.

[Reviewer NfGo]: Added appendix discussing additional related work.

[Reviewer NfGo]: added works on "flat minima"

[Reviewer NfGo]: added prior work on ensemble learning.

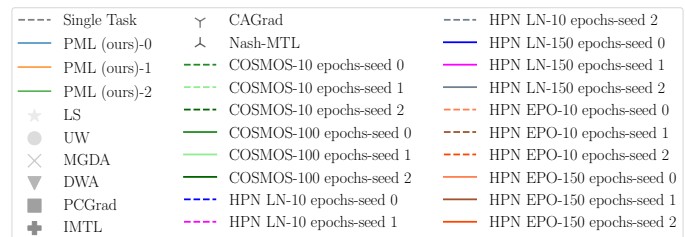

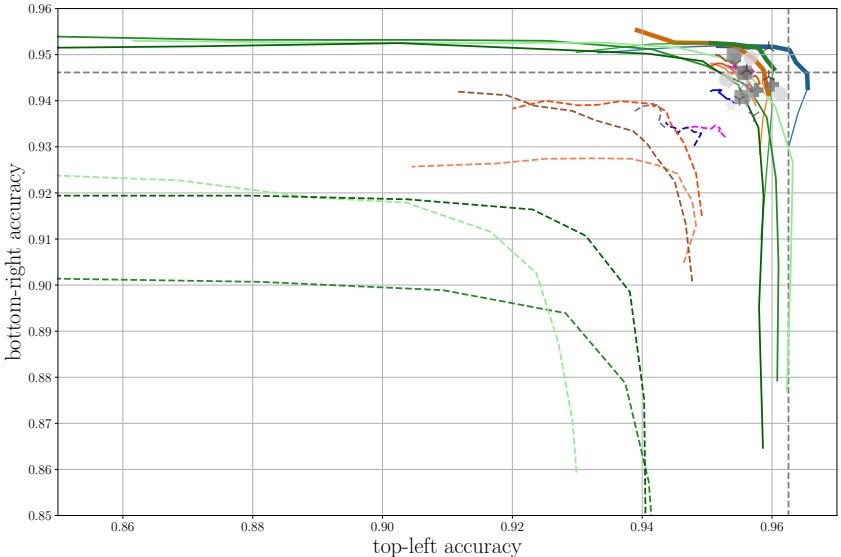

Figure 29: `MultiMNIST`: Figure 4 with additional baselines.

## K ADDITIONAL EXPERIMENTS

In this section, we supplement our experimental findings on `MultiMNIST` with additional baselines, namely HPN-LN and HPN-EPO (Navon et al., 2021) and COSMOS (Ruchte & Grabocka, 2021)[1]. We use hyperparameters of Ruchte & Grabocka (2021) for both methods. We provide two experimental settings:

- *Setting I*: 10 epochs and no learning rate scheduler, i.e., the setting used for all other methods in Figure 4,

- *Setting II*:the experimental setting used by (Ruchte & Grabocka, 2021), i.e., 100 epochs for COSMOS and 150 epochs for HPN-LN/HPN-EPO with multi-step learning scheduler.

Figure 29 presents the results with the additional baselines, using three seeds each. We use dashed lines for *setting I* and solid lines for *setting II* and group the three methods in various color shades (blue, green, red) for visual clarity. We observe that in the original setting of 10 epochs, all new baselines are suboptimal compared to all methodologies. For *setting II*, the hypernetwork methodologies are competitive with some baselines but are suboptimal compared to the proposed method. For COSMOS, only one seed is competitive with the proposed method. Moreover, HPN-LN, HPN-EPO employ a hypernetwork of 1.6m parameters, while the target network has $< 50$k parameters.

[All reviewers]: Added appendix with additional baselines.

---

[1]We use the open source implementation provided by Ruchte & Grabocka (2021) making minimal changes. Our implementation of the `MultiMNIST` dataset has images of size $28 \times 28$ rather than $36 \times 36$ resulting in slightly different models.

