# OpenReview forum: "Pareto Manifold Learning: Tackling multiple tasks via ensembles of single-task models"
_ICLR.cc/2023/Conference — Submitted to ICLR 2023_

### Official Review · Reviewer_NfGo · 2022-10-25

**Confidence:** 4
**Correctness:** 2
**Technical Novelty And Significance:** 2
**Empirical Novelty And Significance:** 2
**Recommendation:** 6

**Clarity, Quality, Novelty And Reproducibility:**

**Clarity:** This paper is well-written and easy to follow.

**Quality:** The paper has a good overall quality, but there are also some major concerns on its main contributions as listed in the weaknesses.

**Novelty:** Some very closely related works are not discussed or compared in this paper, which makes it very hard to judge the novelty and actual contribution of this work.

**Reproducibility:** The proposed model structure seems simple and straightforward. But the experimental results could be not robust, given 1) the current findings on MTL with multi-objective optimization, and 2) the unsolid experimental setting as discussed in the weaknesses.

**Strength And Weaknesses:**

**Strengths:**

+ This paper is well-written and easy to follow.

+ Multi-task learning and multi-objective optimization are both important research topics, and this work tries to further bridge these two fields.

+ The proposed method has reasonable performance on different MTL problems (but with many concerns listed below).

**Weaknesses**

I have some major concerns on both the two claimed advantages of the proposed method, namely, a) learning the Pareto front for multi-task learning and b) better generalization performance.

**1. Contribution and Closely Related Works**

This work proposes to learn the Pareto front for a multi-task learning problem. However, similar Pareto front learning approaches have already been proposed for multi-task learning [1,2,3,4]. Those methods all allow practitioners to choose their preferred solutions at inference time. All these closely related works are not discussed or compared in this paper. The proposed linear interpolation approach could be similar to using a simple linear hypernetwork in [1,2].

Without a thorough discussion and comparison with these closely related works, it is very hard to judge the novelty and actual contribution of the proposed PML method.

**2. Pareto Solution for Better Generalization Performance**

The motivation to learn the Pareto solution for better generalization performance is not strong. Recent works [5, 6] on multi-task learning have shown that a single fixed solution found by simple linear scalarization can have similar or even better performance than those found by multi-objective optimization methods. The effect of careful hyperparameters finetune rather than the Pareto properties could be crucial for good generalization performance.

In addition, the experimental results on the found Pareto front and better per-task performance are somehow not solid. The major issue is that, for an MTL problem with m tasks, the proposed PML model is m times larger than the other baselines, which could lead to unfair comparison. For example, in Figure 4, some baseline models can have comparable or even better performance than the PML's Pareto front. If we can double their model capacity to match the one for PML, it could be likely that some of them (a single solution) can totally dominate the whole Pareto front by PML. In this case, the "Pareto front" could be useless since all tradeoffs are suboptimal and dominated.

Similarly, in Table 1, the overall per-task performance of PML is already nondominated with other MTL baselines. If the model capacity of all the baselines can be doubled, their performance could be significantly improved. On the other hand, the single-task learning baseline (STL) indeed has the same model capacity with PML. In this problem, it seems that simply having separate models for each task already significantly outperforms PML.

**3. Ensemble Learning Method**

Since the proposed PML method requires learning multiple models and can be treated as an ensemble method, it should be compared with the closely related ensemble learning methods. A naive baseline could be to train m models with the same fixed weight among tasks (e.g., simple LS) or other multi-objective optimization (e.g., MGDA, CAGrad), and then a simple ensemble learning method [7] can be applied to achieve better overall MTL performance. An ensemble of models with different optimization methods, such as LS with different weights or MGDA + CAGrad, could be another choice.

The recent work on ensemble learning could have a smaller model size or faster inference time [8,9], and the weight average methods [10, 11] are also strong alternatives with a single final model. Some related works on mode connectivity have been briefly discussed but not compared in the paper. What is the advantage of the proposed PML model for generalization over those ensemble learning or weight average methods?

**4. Other Flat Minima Method**

One main motivation of this work is the connection between low-loss subspace (e.g., flat minima valley) and better generalization performance. From this viewpoint, it is interesting to know its relation to the entropy-sgd [12] and sharpness-aware minimization methods [13, 14, 15]. Can we simply use these methods to train an MTL model with fixed weights for each task? What is the advantage of PML over those methods?

**Other Questions**

1. In Algorithm 1, should the input size of \theta be m rather than m?

[1] Learning the Pareto Front with Hypernetworks. ICLR 2021.

[2] Controllable Pareto Multi-Task Learning. arXiv:2010.06313.

[3] Scalable Pareto Front Approximation for Deep Multi-Objective Learning. ICDM 2021.

[4] Controllable Dynamic Multi-Task Architectures. CVPR 2022.

[5] In Defense of the Unitary Scalarization for Deep Multi-Task Learning. arXiv:2201.04122, 2022.

[6] Do Current Multi-Task Optimization Methods in Deep Learning Even Help? arXiv:2209.11379, 2022.

[7] Simple and scalable predictive uncertainty estimation using deep ensembles. NeurIPS 2017.

[8] BatchEnsemble: an alternative approach to efficient ensemble and lifelong learning. ICLR 2020.

[9] Training independent subnetworks for robust prediction. ICLR 2021.

[10] Averaging Weights Leads to Wider Optima and Better Generalization. UAI 2018.

[11] Model soups: averaging weights of multiple fine-tuned models improves accuracy without increasing inference time. ICML 2022.

[12] Entropy-SGD: Biasing gradient descent into wide valleys. ICLR 2017.

[13] Sharp Minima Can Generalize For Deep Nets. ICML 2017.

[14] Fantastic generalization measures and where to find them. ICLR 2020.

[15] Sharpness-Aware Minimization for Efficiently Improving Generalization. ICLR 2021.


**Summary Of The Paper:**

This work proposes a Pareto manifold learning approach to produce a continuous Pareto front for a given multi-task learning (MTL) problem in a single run. The MTL problem is formulated as a multi-objective optimization problem, and the proposed method can be treated as an ensemble approach for multiple single-task models with linear weight interpolation. In the proposed method, each task (objective) has its own neural network model (with identical structure), and their linear weight combination will be explicitly associated with the corresponding trade-off losses (with linear scalarization). Experimental results show that this method can successfully find the Pareto subspaces for different MTL problems and outperforms other MTL methods.

**Summary Of The Review:**

This paper is well-written, and it studies an important research topic that could further bridge the field of multi-task learning and multi-objective optimization. I personally enjoy reading this work. However, due to the major concerns on main contributions, relation to closely related work, and the experimental setting, it is hard for me to vote to accept the current manuscript.

---

> ### Author Response · Authors · 2022-11-16
> **Response to Reviewer NfGo**
>
> (Part 1/2)
>
> We thank reviewer NfGo for their detailed and very constructive feedback and for bringing our attention to a great list of references. We appreciate that they believe our work is tackling an important research topic and they deem the manuscript is well written.  First, we would like to redirect to [our general message to all reviewers](
> https://openreview.net/forum?id=C9uEwyfklBE&noteId=aVlVQ6dUft
> ) where we address “weakness 1” regarding contribution and closely related works. We now expand on the other comments section by section. We hope that the details provided below will alleviate the concerns delineated in the “summary of the review”, regarding closely related work (see point 1 below) and experimental setting (see point 2 below).
> In general, we have cited all prior works mentioned in the review, either in the main text or in the appendix J (page 40) due to space limitations.
>
> For ease of reading we refer to the references of the review by appending the letter r (e.g. [r2]). We cite more references below appending the letter p (e.g. [p3]).
>
> **1. Contribution and Closely Related Works**
>
> We have revised the manuscript to showcase the differences of our proposed method with these prior works and included appendix K (page 41) where we expand the list of baselines for MultiMNIST to include references [r1,r3]. We compare and contrast with [r1] in our general reply to all reviewers, since this prior work was also mentioned by other reviewers. The same comments apply to [r2], which is concurrent to [r1] and also uses hypernetworks. [r4] also utilizes hypernetworks, but is better classified as a Neural Architecture Search method since it employs a dynamic architecture, while our architecture is fixed. Also, a major difference is that [r4] employs pretrained models for each task, while we do our training in a single run. [r3] augments the input by the preference vector and trains a single model. On the other hand, we employ a (weight) ensemble of models.
>
> **2. Pareto Solution for Better Generalization Performance**
>
> Both of these concurrent works [r5, r6] are very interesting. Thank you for bringing them to our attention. However, please note that as they will appear at NeurIPS 2022 which will take place after this rebuttal, they are considered as “contemporaneous” works as per the [ICLR conference guidelines](https://iclr.cc/Conferences/2023/ReviewerGuide#faq).
>
> In any case, we believe that these papers highlight strengths of our paper rather than weaknesses, a point we also raised in the [general reply to all reviewers](https://openreview.net/forum?id=C9uEwyfklBE&noteId=aVlVQ6dUft). Specifically, while baselines might not dominate Linear Scalarization, as argued in [r5,r6] and also shown in, e.g., Figure 4 of our paper, our method is able to incorporate the tradeoffs and find a subspace that can accommodate all in a single training run.
>
>
> We also believe that our experimental settings are robust and solid. Regarding the results of Figure 4, for Census all solutions dominate the baselines and for multiMNIST all seeds either completely dominate baselines (e.g. seed 0) or consistently outperform them. During training, we indeed have m times the parameters as any of the baselines. However, this excess of parameters (to a higher or lesser degree) is prevalent in all prior works which result in multiple solutions such as [r1,r2,r3,r4,p1]. During inference all methods have exactly the same architecture and number of parameters (except for Single-task which has one head and, therefore, slightly fewer parameters). Specifically, assume that a multi-task predictor has $p_{shared} +m\cdotp_{task} \approx p_{shared} =p$  parameters in total. Each ensemble member has the same amount of parameters. This means that we train with $mp$ parameters. For Figure 4, the ensemble members are connected by a linear segment, which is sampled 11 times and then a model (with $p$ parameters) is accordingly formed via a convex combination. We report the results of all these 11 models pictorially in Figure 4.

---

> > ### Author Response · Authors · 2022-11-16
> > **Response to Reviewer NfGo (part 2)**
> >
> > (Part 2/2)
> >
> > **3. Ensemble Learning Method**
> >
> > All of these methods share some traits with our proposed method but are quite different. For example, all concern single-task learning. Still, the connection is valuable and we have added references to all in our related work section, discussing their differences with our approach. Let $m$ be the number of ensembles. With more details:
> > * [r7] produces an output ensemble while we focus on weight ensembles. This means that inference costs m times more. It is also unclear how one can infuse Pareto properties to such a setup since [r5,r6] show that most baselines yield similar solutions in objective space (see [r6, Fig 2]).
> > * [r8] does improve on the computational overhead of [r7] but still presents hurdles in its extension to Multi-Task Learning. For instance, it assumes an m times larger batch size [r8, section 3.2] which might be limiting in some cases.
> > * [r9] accommodates multiple members in a single model by assigning different subset weights to each. This means that no weight ensemble can be formed since each member has a different structure. Effectively, a discrete set of (incompatible) predictors cannot approximate a continuous Pareto Front. Moreover, this leads to lower representational capacity on individual members [r9, Fig 5].
> > * [r10] produces one model in the “middle” of the basin and, consequently, encodes only one set of tradeoffs (similarly to any MTL baselines explored in our paper). We experimentally show a different perspective outlined in our general reply to all reviewers. We have also added a new appendix called “Connection between Pareto Optimality and multiple valley intersections” that explores this novel view.
> > * [r11] focuses on fine-tuning of large models in order to produce a weight ensemble. This means that (a) [r11] requires heavy pretraining (b) the models are already in the same basin and averaging will work as per observations in [r11, p2].
> >
> > Overall, our method differs from all the aforementioned works and proposes a novel view of multi-task learning as a collection of multiple loss landscapes and satisfies the desideratum of continuous approximation of the Pareto Front. We have included these references in the appendix due to space limitations.
> >
> > **4. Other Flat Minima Method**
> >
> > The discussed works [r12,r13,r14,r15] focus on single-task learning. In principle, each of these methods can be used for a fixed-weight linearly-scalarized multi-task objective. This is an interesting research question. However, this will result in a single point/solution with predefined trade-offs which is qualitatively different from our goal of Pareto diversity, i.e., to produce a continuous and diverse solution set. We have included these references in the appendix due to space limitations.
> >
> >
> >
> > **5. Other questions**
> >
> > The input size of $\Theta$ can be either $T$ (number of tasks) or m (number of ensemble members) since for simplicity we have assumed that $T=m$.
> >
> > [p1] Pareto Multi-Task Learning. NeurIPS 2022
> >
> > [p2] Linear Mode Connectivity and the Lottery Ticket Hypothesis. In ICML 2020.

---

> ### Comment · Reviewer_NfGo · 2022-11-22
> **Thank You and Follow-up Comments**
>
> Thank you for your detailed response. Here are my follow-up comments.
>
> **Contribution**
>
> Technically, I still believe the proposed PML weight ensemble is equal to a simple linear hynernetwork. You can always use a larger hypernetwork for more model capacity, keep using the most simple linear hypernetwork without other design choices (as in this work), or use the chunking method to further reduce the model size. It is not suitable to say fixing the choice as the linear hypernetwork and banning other design choices is an advantage of the proposed method. Therefore the technical novelty over the existing Pareto front learning method is not significant.
>
> **Pareto Solution for Better Generalization Performance**
>
> Regarding the diverse tradeoffs, the concern is on the more realistic problem with a larger deep neural network model and different loss functions rather than the toy Census and MultiMNIST problems. For a real-world deep (multi-task) learning problem, the solution with the lowest training loss value usually does not have the best testing performance (e.g., accuracy) due to generalization and overfitting. There is a generalization gap from training loss -> testing loss -> testing performance, and it seems that a Pareto solution on the training loss is not necessarily a Pareto solution on the testing performance that we care about. Will the idea of the Pareto subspace provide any insight on this generalization gap? Could the proposed PML method find a meaningful Pareto front for the CityScapes problem?
>
> In Table 1, are the results for PML (e.g., Segmentation and Depth) from the same single solution (and which one)? From the current Section 5.3, it seems that the results are from different solutions for each task separately. If this is the case, PML still has m times parameters for both training and inference.
>
> **Ensemble Learning and Other Flat Minima Method**
>
> Thank you for the detailed discussion.
>
> The major concern is that the flat minima methods and some ensemble learning methods, with the same or even smaller parameters with PML (for training/inference), can find a better solution than the MTL baselines. If their solutions can dominate the Pareto front by PML, the found "Pareto front" could be useless since all tradeoffs are suboptimal. Therefore, it could be interesting to see their comparison on more realistic problems with deep neural network models (such as the CityScapes problem).

---

> > ### Author Response · Authors · 2022-11-24
> > **Response to Follow-up Comments by Reviewer NfGo**
> >
> > (Part 1/2)
> >
> > Thank you for your follow-up discussion. We reply per section below:
> >
> > **Contribution**
> >
> > We disagree regarding the significance of the contribution. Overall, we believe that casting a weight ensemble as a hypernetwork is a vacuous generalization that downgrades the "generation" of weights to "storing" of weights and invalidates weight ensemble methodologies. It could be similarly argued that MLPs subsume all network architecture; CNNs can be seen as special cases of MLPs. We expand on our reasoning below. Please check our [general message to all reviewers](https://openreview.net/forum?id=C9uEwyfklBE&noteId=aVlVQ6dUft) for an initial discussion.
> >
> > 1. **hypernetworks are qualitatively different to weight ensembles**: hypernetworks are used to generate the weights of another (target) network. This means that the target network is produced via *complex interactions* of, e.g., a MLP. On the other hand, weight ensembles simply interpolate in the complex hull members' weights. This qualitative difference can also be (implicitly) seen by papers on weight ensembles [2,3,4] which do not even mention hypernetworks. Mathematically, let $h(r; \phi)$ be the hypernetwork function that produces weights  $\theta$, then used by the target network, for prediction via the function $f(\cdot; \theta)$. Mathematically, a MLP hypernetwork has parameters $\lbrace\mathbf{W}^L,\cdots,\mathbf{W}^1, b_L,\cdots,b_1\rbrace$ produces weights via the function $\theta=\sigma(\mathbf{W}^{L}(\sigma(\mathbf{W}^{L-1}(\cdots \sigma(\mathbf{W}^1\mathbf{a})\cdots) + b_L)))$. On the other hand, our method has parameters $\lbrace\theta_1,\theta_2\rbrace$ produces the weights $\theta=a\theta_{1}+ (1-\alpha)\theta_2$.
> > 2. **Inductive bias and geometrical insights**: Even if all weight ensembles can be seen as particular cases of hypernetworks, **an important contribution of our paper is the existence of Pareto Subspaces.** We provide the geometrical insight of casting the multi-task problem as finding the intersection of multiple loss landscapes and, going one step further, that the transition from focusing on one task to the other can be done via simple subspaces. **This geometrical conjecture leads to weight ensembles** since the relationship is *explicitly* set via the convex combination of member parameters and loss weights. But, this important connection and geometrical insight are lost in hypernetworks which generate weights via this complex mapping $\theta=\sigma(\mathbf{W}^{L}(\sigma(\mathbf{W}^{L-1}(\cdots \sigma(\mathbf{W}^1\mathbf{a})\cdots) + b_L)$.
> > 3. **Performance**: In appendix K we have shown that this qualitative difference translates to difference performance for hypernetworks and weight ensembles. This is similar to the comparison of MLPs and CNNs; while CNNs are a special case of MLPs their performance is significantly better (and usually achieved with less training) on vision tasks. By the same token, our method trains faster and outperforms hypernetworks.
> > 4. **Simplicity is important**: As argued in points (1) and (2), the choice of weight ensemble arises naturally from the our geometrical intuition and did not arise from the connection to hypernetworks, i.e. it did not come via "banning other design choices". We believe that this is an advantage that actually helps the model's performance and convergence (see appendix K). For example, ResNets fix some architectural aspects by the residual connection which theoretically does not change something (i.e. the block can still learn the same mapping) but is crucial in practice.

---

> > > ### Author Response · Authors · 2022-11-24
> > > **Response to Follow-up Comments by Reviewer NfGo (continued)**
> > >
> > > (Part 2/2)
> > >
> > > **Pareto Solution for Better Generalization Performance**
> > >
> > > Regarding the tradeoffs and the experimental setup, we do agree that there is a generalization gap from training loss -> testing loss -> testing performance, and this is why we actually report testing performance. **This is actually a strong point of our experimental setup compared to prior works [1,5] on two points**. First, we actually compare with SOTA Multi-Task methods and are either competitive or outperforming them while prior works include comparison with each other and simple baselines such as uniform weights. Second, we report test performance (e.g. test *accuracy*) instead of potentially misleading metrics such as test *loss* and/or HyperVolume.
> > >
> > > - *Will the idea of the Pareto subspace provide any insight on this generalization gap?* This is an important question that still has not been answered for single-task learning. Learning multiple tasks compounds on the complexity of this question and, hence, is best suited for future work.
> > > - *Could the proposed PML method find a meaningful Pareto front for the CityScapes problem?*: We opted for a tabular presentation for Cityscapes dataset in order to be compatible with prior work that also report 4 metrics. Our experiments showed that the Pareto Front is narrower and concentrated in the "middle" of the line, e.g. $\alpha\in[0.3, 0.7]$, but is still meaningful. It is important to note that the results are obtained via sampling uniformly 11 times across this line (i.e. $\alpha=0, 0.1, 0.2,\cdots, 0.9, 1$) which actually show a lower estimate of our method's performance.
> > >
> > > **Ensemble Learning and Other Flat Minima Method**
> > >
> > > This is a very interesting line of research and are best suited for future research. However, it is beyond the scope of this paper, since we focus on the objective of continuous approximation of the Pareto Front rather than point estimates.
> > >
> > > [1] Navon A, Shamsian A, Fetaya E, et al. Learning the Pareto Front with Hypernetworks ICLR  2020.
> > >
> > > [2] Learning Neural Network Subspaces ICML 2021.
> > >
> > > [3] Model soups: averaging weights of multiple fine-tuned models improves accuracy without increasing inference time. ICML 2022.
> > >
> > > [4] Averaging Weights Leads to Wider Optima and Better Generalization, UAI 2018
> > >
> > > [5] Scalable Pareto Front Approximation for Deep Multi-Objective Learning. ICDM 2021.

---

> > > > ### Comment · Reviewer_NfGo · 2022-11-25
> > > > **Thank you for two rounds of thorough response**
> > > >
> > > > Thank you for your two rounds of thorough responses. I now feel very broaderline about this work. I raise my score to 6 and encourage the author to continually investigate this topic further and deeper.
> > > >
> > > > **Contribution and Simplicity**
> > > >
> > > > I respectfully disagree with the "weight ensemble v.s. hypernetwork" to "CNN v.s. MLP" comparison. CNN is a carefully and specially designed neural network that leverages the shift equivariant property of the input data (e.g., an image) for efficient modeling. But this is not the case for a simple linear hypernetwork. In the MLP hypernetwork, if we simply set $W^L = \ldots = W^2 = I$ and use identity activation, we have $\theta = W^1a$ which is the weight ensembles of $W^1 = [\theta_1, \theta_2]$. This structure does not leverage special invariant/equivariant properties. Therefore, I believe the "simplification" (e.g., do not use MLP and chunking for better scalability) is not a novel structure as for CNN over MLP.
> > > >
> > > > In addition, without mentioning hypernetwork, the simple linear weight ensemble has also been applied to get different middle (e.g., trade-off) models for continuous imagery effect transition [1]. But I believe the proposed PML has its own advantages over [1], such as training the whole Pareto front in a single run rather than simple linear interpolation with multiple trained models (their middle models could have high losses).
> > > >
> > > > [1] Deep Network Interpolation for Continuous Imagery Effect Transition. CVPR 2019.
> > > >
> > > >
> > > > **Pareto Solution and Better Generalization**
> > > >
> > > > I think the generalization gap could be a bit more important for PML than the single-task counterpart. Especially for a deep neural network, a found continuous Pareto front on the training loss is indeed not a Pareto front on the testing accuracy. In this case, there is actually no guarantee on the solution's performance or its Pareto property. For example, for solution A found by weight [0.7, 0.3] and solution B found by weight [0.3, 0.7], a practitioner could expect that solution A has better performance on task 1 and solution B has better performance on task 2, and they should be both Pareto solutions. However, due to the generalization gap, there is no such guarantee. Solution A can either 1) totally dominates solution B, 2) be totally dominated by solution B, or 3) have better performance on Task 2 but not 1... It is also possible that both A and B indeed are not Pareto solutions. In this case, the found continuous Pareto front (Pareto subspace) could be much less promising than originally expected as in the second contribution that "enabling prac75 titioners to handpick during inference the solution that offers the tradeoff that best suits their needs".
> > > >
> > > > For the CityScapes experiment, does it mean that the PML first obtains 11 solutions with uniform weights, and then reports the best results for each task among the 11 found solutions? If this is the case, it means PML needs 2x parameters and 11x inference budgets to find solution that are non-dominated with other MTL baselines. Why not simply report the results for different weights such as {[1,0], [0.5,0.5], [0,1]} (or all 11 solutions)?
> > > >
> > > > **Comparison to Other Flat Minima Methods**
> > > >
> > > > I actually do agree and appreciate the author's claim in the introdutction "(for the geometry of the loss landscape and linearly connected local optima), when the problem has multiple objectives, it acquires a new dimension relating to the number of tasks." And I think the proposed PML could be a good contribution to this direction.
> > > >
> > > > However, the advantage of this new dimension (e.g., Pareto subspace) over other flat minima methods is not properly explored in this work. For example, the first claimed advantage of PML (in addition to the Pareto front) is "enforcing low loss for all tasks on a linear subspace implicitly penalizes curvature, which has been linked to generalization, benefitting all tasks' performance". However, it is unclear whether PML can outperform other flat minima /ensemble learning methods that train an MTL model with fixed weights for better generalization. Although PML can find a continuous approximation of the Pareto front, the Pareto front could be useless if it is dominated by a single solution found by other flat minima methods.

---

> > > > > ### Author Response · Authors · 2022-11-27
> > > > > **Thank you and response**
> > > > >
> > > > > Thank you for raising your score, the very constructive comments and fruitful discussion. We reply per section below.
> > > > >
> > > > > **Contribution and Simplicity**
> > > > >
> > > > > Thank you for pointing out the advantages of PML. Indeed, our focus lies on multiple tasks and the proposed algorithm is suitable towards obtaining Pareto solutions. Compared to [1], training occurs from scratch rather than as a secondary phase, i.e., fine-tuning.
> > > > >
> > > > > **Pareto Solution and Better Generalization**
> > > > >
> > > > > It is indeed theoretically possible for the described situation to arise, i.e., for solution A to have lower performance than solution B for task 1 and vice versa for task 2, while we expect the opposite. However, in practice, the monotonic relationship between the “interpolation” and “task” weights implicitly prevents this phenomenon and leads to the desired Pareto property. This alignment was observed throughout training.
> > > > >
> > > > > For the CityScapes experiment, we train one ensemble with two models, which correspond to the single-task weights $[1, 0]$ and $[0, 1]$. The solution obtained by PML is the subspace corresponding to the line segment between the models. For evaluation, e.g., plots in Figure 4, we sample the subspace 11 times and measure the performance by the respective models, which correspond to the interpolation weights $\lbrace [0,1], [0.1, 0.9], [0.2, 0.8] , [0.3, 0.7],\cdots, [0.9, 0.1], [1., 0]\rbrace$. For Cityscapes, this presentation would lead to a large table and, thus, we opted to “present the best-of-(sampled)-subspace results”.
> > > > >
> > > > > **Comparison to Other Flat Minima Methods**
> > > > >
> > > > > Thank you for your comment regarding the geometrical insight. The new dimension (number of tasks) is a challenge rather than an advantage. To be more specific, the *per-task* loss landscape must remain flat throughout the solution subspace so that all (interpolated) solutions achieve low loss/high performance. The “Pareto subspace property” means that for the *weighted (multi-task)* loss landscape, the models lying in the solution subspace are arranged (w.r.t. weight $\alpha$) in an increasing performance sequence for one task and a decreasing for the other. Hence, the subspace contains Pareto solutions. In this perspective, we do not compare/compete with other flat minima methods but rather state that a benefit of the algorithm is implicitly finding flat minima, both in the "static" sense of *per-task* loss landscapes and in the novel view of the *dynamic transition* of the weighted loss landscape.
> > > > >
> > > > > [1] Deep Network Interpolation for Continuous Imagery Effect Transition. CVPR 2019.

---

### Official Review · Reviewer_qexX · 2022-10-26

**Confidence:** 3
**Clarity, Quality, Novelty And Reproducibility:** This paper is well organized and easy…
**Correctness:** 3
**Technical Novelty And Significance:** 3
**Empirical Novelty And Significance:** 3
**Recommendation:** 5

**Strength And Weaknesses:**

Strengths:

1. This paper improves the method in Ma et al. (2020) and proposes a novel method to produce continuous Pareto stationary points, which is simple and easy to utilize. In addition, the method is efficient as it requires only a single run.
2. This paper is inspired by the theory of single-task machine learning and the motivation is quite interesting. Meanwhile, the concept of Pareto manifold learning is insightful and worth exploring.
3. The visual explanation provided in Figure 3 is easy to understand, while the visual analysis of the article (Figure 4 and Figure 5) is comprehensive.

Weaknesses:

1. It must be acknowledged that the conjecture about the loss landscape from the single-task to multi-task is insightful. However, it is better to give a toy example to explain it. More importantly, even though multiple valley intersections in the low loss region in the multi-task scenario, this observation is not strongly related to Pareto manifold learning. I suggest the author should explain it more clearly.
2. In the last paragraph of the introduction, the author announces that “the algorithm produces a subspace of Pareto Optimal solutions”, It should be pointed out that multi-task learning methods can only ensure convergence to Pareto optimal when assuming it holds a convex loss, in other words, we can only approximate rather than obtain the Pareto optimal solutions.
3. The proposed method fails to exceed the baseline methods such as Nash-MTL and PCGrad on several datasets(Table 1, Table 5, and Table 6), which makes the method less convincing. In addition, the ablation experiment demonstrated in Tabel 2 takes a value of $\lambda$ equal to 0 on the MultiMNIST dataset, which may indicate that the regularization is not effective. Most importantly, there is a lack of detailed analysis of the experimental results.
4. Paper writing should be polished, For instance, Figure 1 and Figure 2 are not mentioned in the context, which is not friendly to read.
5. In addition, some typos exist in the paper. In line 11 of Algorithm 1, the citation of Figure 4.2 is wrong, the same problem is found in the corresponding text above Claim 3, I guess it should be Figure 3.


**Summary Of The Paper:**

This paper proposes a novel approach to seek a continuous Pareto Front for multi-task learning. The key idea is to train multiple single-task predictors and performs convex hull operations on different models in the weight space to produce a subspace with multiple Pareto stationary points. A continuous Pareto Front can be produced through only a single run in this method.

**Summary Of The Review:**

This paper focus on an interesting problem and provides some new ideas. However, there are some technical flaws in this work.

---

> ### Author Response · Authors · 2022-11-16
> **Response to Reviewer qexX**
>
> We thank the reviewer qexX for their comments. We appreciate that they find our work insightful and interesting. First, note our global comments in our [general message to all reviewers](https://openreview.net/forum?id=C9uEwyfklBE&noteId=aVlVQ6dUft). We expand on each comment individually:
>
> **Weakness 1.**
>
> We are pleased that the reviewer deems our conjecture insightful. In order to clarify the connection between the intersection of low-loss regions and Pareto Subspaces/Pareto Manifold learning, we have added a new appendix called “Connection between Pareto Optimality and multiple valley intersections” (appendix I, page 37) where we provide a toy example and visualizations.
>
> **Weakness 2.**
>
> This distinction has been addressed in page 5 to reflect the fact that in the non-convex setting of Neural Networks global optimality cannot be guaranteed and, instead, we relax this desideratum to local optimality, e.g. Pareto optimality in some open neighborhood. This distinction is standard in the literature, e.g. [1].
>
> **Weakness 3.**
>
> We are confused regarding the reviewer’s comment about performance comparison with the baselines. Overall, in Tables 1, 5 and 6 there is only one table (Table 5) where two multi-task baselines beat (LS, CAGrad) our method. We address each table separately below. We understand that the naming of baselines in the appendix is not clear and we will change the tables/captions to better reflect this. To further improve legibility, we have added bold font to tables 5 and 6.
> * Table 1: No Multitask baseline outperforms our proposed method. For example, PCGrad (one of the methods mentioned in the review) performs considerably worse in the task of Depth (e.g. Abs err metric: 0.0140 vs 0.0185, lower is better). Overall, PML (ours) dominates all multi-task baselines in the task of Depth. The only exception is MGDA, but PML outperforms it on the other task of semantic segmentation. We have updated “Section 5.3 Scene Understanding” to mention that no MTL method dominates the proposed method.
> * Table 5: STL refers to Single-Task Learning, which is the only baseline that dominates our method. On the other hand, our method outperforms all Multi-Task baselines on all tasks. We have added bold font for the best-per-task and a horizontal line to differentiate between single-task and multi-task baselines.
> * Table 6:  We have added bold font for the best-per-task and a horizontal line to differentiate between single-task and multi-task baselines. CAGrad produces the best performance but the proposed method performs better on other benchmarks (e.g. Figure 4). Please note that the results on UTKFace do not utilize the full method, i.e., no multiforward, but rather ablate the importance of balancing schemes in order to show that they can be effective in cases, for instance, where tasks have different loss scales.
>
> Regarding the ablation experiment for Table 2 where the optimal value of $\lambda$  is 0. We have added a comment on Appendix D addressing this. Please note that multiforward training (and not only the regularization) does help since all optimal HV values (per and across seeds) are achieved for window $W=4$. The fact that the optimal value of  is zero can be attributed to the symmetric nature of the MultiMNIST benchmark.
>
>
> **Weakness 4.**:
>
> Thank you for drawing our attention to this. We have updated the manuscript accordingly.
>
> **Weakness 5.**
>
> Thank you for drawing our attention to these typos. We have updated the source code to remedy them.
>
>
>
> [1] Multi-Task Learning as Multi-Objective Optimization. NeurIPS 2018

---

### Official Review · Reviewer_SqFR · 2022-11-03

**Confidence:** 4
**Correctness:** 3
**Technical Novelty And Significance:** 3
**Empirical Novelty And Significance:** 3
**Recommendation:** 5

**Clarity, Quality, Novelty And Reproducibility:**

Quality:
This paper is well written and easy to read. The organization is clear.

Clarity:
Some issue of motivations and method details need more clarification. Please see weakness section.

Originality:
fair.


**Strength And Weaknesses:**

Strength:
This paper castes multi-task problems as learning an ensemble of single-task predictors by interpolating among members during training, and designs weight subspaces where multiple optimal functional solutions lie. Based on this weight space, an ensembling method that can produce a continuous Pareto Front in a single training run is proposed. This allows practitioners to modulate the performance on each task during inference on the fly.

Weaknesses:

1.Similar methods have already been proposed for multi-task learning and has not been disccussed in this paper [1].

1.When sampling on the convex hull parameterization, authors choose to adopt the Dirichlet distribution since its support is the T-dimensional simplex. Does this distribution have other properties.   Why using this distribution?  If p≫1，how the ensemble will change.

2.When training, a mono tonic relationship is imposed between the degree of a single-task predictor participation and the weight of the corresponding task loss. As a result, the ensemble engenders a subspace that explicitly encodes tradeoffs and results in a continuous parameterization of the Pareto Front.  Whether the mono tonic relationship can be replaced by other relationships? Explaining this point may be better.

[1]Navon A, Shamsian A, Fetaya E, et al. Learning the Pareto Front with Hypernetworks[C]//International Conference on Learning Representations. 2020.

**Summary Of The Paper:**

This paper assumes there exist Pareto Subspaces, i.e., weight subspaces where multiple optimal functional solutions lie, and develops a weight-ensembling method named Pareto Manifold Learning that casts multi-task problems as learning an ensemble of single-task predictors by interpolating among members during training.  Multiple single-task predictors are trained in conjunction to produce a subspace formed by their convex hull, and endowed with desirable Pareto properties. Each single-task model infuses and benefits from representational knowledge and the other members. The losses are weighted in tandem with the interpolation when training.  The ensemble as a whole engenders a weight subspace that explicitly encodes tradeoffs and results in a continuous parameterization of the Pareto Front.

**Summary Of The Review:**

This paper produces a subspace with multiple Pareto stationary points in the multi-task learning based on the geometry of the loss landscape in single task machine learning where the local optimal are connected by simple paths. However, the representation and more illustrations are needed.

---

> ### Author Response · Authors · 2022-11-16
> **Response to Reviewer SqFR**
>
> We thank reviewer SqFR for their valuable and constructive feedback. First, we would like to redirect to [our general message](https://openreview.net/forum?id=C9uEwyfklBE&noteId=aVlVQ6dUft) where we reiterate strengths of the proposed method and address some global comments. We now proceed to address the rest of the comments:
>
> **Weakness 1: Prior work**
>
> We have revised the manuscript to address this concern. We have also included a discussion regarding the paper mentioned in [our general message to all reviewers](https://openreview.net/forum?id=C9uEwyfklBE&noteId=aVlVQ6dUft).
>
> **Weakness 2: Sampling with Dirichlet distribution**
>
> We have updated the manuscript and added a new appendix (appendix H in page 35) to explore more deeply the connection between sampling and final performance. Indeed, our selection of the Dirichlet distribution $\textrm{Dir}(\mathbf{p})$ stems from the fact that its support is the $T$-dimensional simplex making it a natural choice. Essentially, we want to sample only in the convex hull of the ensemble members (in parameter space) and this particular distribution choice is aligned with this objective. We make a design choice by selecting all concentration parameters to be equal $\mathbf{p}=p\mathbf{1}$, i.e., we place no prior knowledge about task difficulty and favor no task. Effectively, this results in three cases: $p=1$, $p(0,1)$  and $p>1$. We explore the first two cases in the appendix and expand on the third (i.e., the one mentioned in the review) here. Assume an extreme case of two tasks and $p$. Then, the distribution becomes deterministic and outputs equal weights for all tasks. The randomly and independently initialized ensemble members will collapse to each other, resulting in duplicate ensemble members. Similarly, for very large values (e.g. $p=100$), the functional diversity will suffer. This can be seen in the experiment of Figure 25a on page 36.
>
> **Weakness 3: Replacing monotonic relationship by other relationships**
>
> It is not completely clear to us, if  the comment refers to other monotonic, or rather to other non-monotonic relationships? The use of a monotonic relationship is a key insight of our method that, indeed, can benefit from an extended exposition. Specifically, we have two sets of weights: $w_{loss}$ regarding the loss aggregation (as in Linear Scalarization baseline) and $w_{ensemble}$ regarding the weight ensembling. In equation 2 of the paper (and the short paragraph below), both are equal and denoted by $\alpha$.
>
> The reasoning behind this is that the closer the current model is to an ensemble member, the more this should be reflected in the corresponding task loss. This leads to retrieving a Pareto subspace since we have associated one ensemble member to each task. The choice of having setting $w_{ensemble}=w_{loss}=a$ can be relaxed and replaced by a monotonic relationship. Then, the desirable property of associating an ensemble member to each task is maintained.

---

### Author Response · Authors · 2022-11-16
**General reply to all reviewers**

(Part 1/2)

We thank all the reviewers for their valuable and constructive feedback. We are pleased that the reviewers deemed our paper interesting ([qeXX](https://openreview.net/forum?id=C9uEwyfklBE&noteId=fmgT50KMSe)), well written and with easy-to-understand visualizations ([SqFR](https://openreview.net/forum?id=C9uEwyfklBE&noteId=En9sQnN7PEc), [qeXX](https://openreview.net/forum?id=C9uEwyfklBE&noteId=fmgT50KMSe), [NfGo](https://openreview.net/forum?id=C9uEwyfklBE&noteId=hZA9Bbt3Sjr)), simple and efficient ([qeXX](https://openreview.net/forum?id=C9uEwyfklBE&noteId=fmgT50KMSe)), insightful ([qeXX](https://openreview.net/forum?id=C9uEwyfklBE&noteId=fmgT50KMSe)) and addressing important research topics ([NfGo](https://openreview.net/forum?id=C9uEwyfklBE&noteId=hZA9Bbt3Sjr)).

We provide here a general reply for points raised by several reviewers, and also reply individually (in separate OpenReview threads) to each reviewer to address their respective questions. We have uploaded a new version of the paper, where we use red font to show the changes. For ease of navigation we have added “notes” pinpointing the comment and the reviewer each corresponding change addresses.

First, we reiterate our key contributions:
1. We present a novel view of multi-task learning as an intersection of multiple loss landscapes which allows us to leverage interesting findings from single-task learning. We conjecture and show experimentally a novel result: the existence of Pareto subspaces.
2. We propose a novel Multi-Task learning algorithm that incorporates these ideas and discovers in a single training run a continuous parameterization of the Pareto subspace, allowing practitioners to select their preferred tradeoff on-the-fly during inference.
3. Our method is simple, scalable to large networks, and it yields a diverse solution set (e.g., Figure 4 of the paper), especially when applying the proposed “multi-forward batch regularization”.

**Missing related work, such as [1]**

We have included a more detailed overview of past works in our reply to reviewer [NfGo](https://openreview.net/forum?id=C9uEwyfklBE&noteId=hZA9Bbt3Sjr). We have updated the manuscript to include such prior work. We also added appendix K (page 41) with more baselines (such as [1]) for the MultiMNIST benchmark, where we show that our proposed method outperforms these additional baselines.

The task addressed by [1] is indeed related to our work since we share the same goal: a continuous approximation of the Pareto Front in a single-run. Therefore we have incorporated it in the revised paper along with proper discussion.
However, it is important to note that the methodology is different, since the work in [1] employs hypernetworks, while we use weight ensembles. Theoretically, our weight ensemble can possibly be thought of as an extreme case of a hypernetwork, which stores all member parameters and produces a convex combination. This may not be the most effective generalization however.  It has been the core business of deep learning of providing different architectures/optimization schemes without necessarily changing fundamentally the functional space. We build our method upon the conjecture that there exist Pareto subspaces of simple geometry that  justifies the form of the convex combination. Along the same lines, methods such as [8,9] still are insightful and use the specific setting of weight ensembling rather than hypernetworks, which confirm that both settings are certainly valid ones.

Apart from a theoretical perspective, there are key practical considerations as well. First, contrary to our method, the use of hypernetworks introduces additional design choices and hyperparameters. Second, we restate verbatim the comments of [2, section 3.1]: *“We introduce an anchor net F as an alternative approach to model weight generation in dynamic networks for MTL [1]. Previous methods adopt chunking to mitigate the large computation and memory required for generating entire network weights at the expense of limiting the hypernet capacity.”* In other words, during inference and in the case of selecting a tradeoff, the work in [1] requires the generation of a new target network which can be either extremely costly (for vanilla hypernetworks) or less costly but in the expense of lower capacity and, hence, lower representational power of the target network (in case of chunking). On the other hand, our method simply requires an addition of ensemble members’ weights. Finally, our method is scalable to large networks such as SegNet [3], while [1] uses ENet [4], which has ~300k parameters. This concern is also voiced by [5] that states verbatim: *“It is important to highlight that none of the baselines can scale to this experiment. ParetoMTL needs to train one different EfﬁcientNet network for each point on the Pareto front, while PHN-LS and PHN-EPO [1] increase the number of trainable parameters to approximately 100x that of an EfﬁcientNet architecture.”*

---

> ### Author Response · Authors · 2022-11-16
> **General reply to all reviewers (part 2)**
>
> (Part 2/2)
>
>
> Finally, we would like to thank reviewer [NfGo](https://openreview.net/forum?id=C9uEwyfklBE&noteId=hZA9Bbt3Sjr) for bringing the concurrent works [6,7] (accepted recently in NeurIPS 2022) to our attention, as we believe they substantiate our method and findings. On a high level, both papers claim that recent SOTA Multi-task methods (i.e,. our baselines) “do not yield any performance improvements beyond what is achievable via traditional optimization approaches” despite their computational complexity. Specifically, [7, Figure 2] shows that SOTA baselines do not yield diverse solutions and their final objective values are predefined by their optimization scheme (and the randomness in a lesser degree) while Linear Scalarization can yield diverse solutions. Our figure 4, is qualitatively similar to [7, Figure 2]  and we do observe the same behavior by baselines (they form a cluster) but our method achieves a high degree of diversity while requiring a single run.
>
> [1] Navon A, Shamsian A, Fetaya E, et al. Learning the Pareto Front with Hypernetworks ICLR  2020.
>
> [2] Controllable Dynamic Multi-Task Architectures. CVPR 2022.
>
> [3] SegNet: A Deep Convolutional Encoder-Decoder Architecture for Image Segmentation. TPAMI 2017
>
> [4] Enet: A deep neural network architecture for real-time semantic segmentation.
>
> [5] Scalable Pareto Front Approximation for Deep Multi-Objective Learning. ICDM 2021.
>
> [6] In Defense of the Unitary Scalarization for Deep Multi-Task Learning. NeurIPS 2022.
>
> [7] Do Current Multi-Task Optimization Methods in Deep Learning Even Help? NeurIPS 2022.
>
> [8] Learning Neural Network Subspaces ICML 2021.
>
> [9] Model soups: averaging weights of multiple fine-tuned models improves accuracy without increasing inference time. ICML 2022.

---

### Decision · Program_Chairs · 2023-01-20

**Decision:**

Reject

**Justification For Why Not Higher Score:**

The paper is quite interesting in terms of idea and method. However, authors simply missed the closest set of related work (hypernetworks) from the literature in their discussion and experiments. The paper needs a major revision of extensive comparison and additional discussion to be presented.

**Justification For Why Not Lower Score:**

N/A

**Metareview: Summary, Strengths And Weaknesses:**

The paper is within the family of handling multi-task learning as multi-objective optimization family. The main contribution is posing the problem as learning a space of solutions such that this linear space will correspond to a Pareto efficiency set. Since the space is assumed to be linear, its extreme points are learned directly within optimization. The method is later evaluated extensively on various benchmarks. The submission was reviewed by 3 experts and they all shared the same sentiment: The paper is very interesting however both discussion and experimental study misses a very significant set of works, hypernetworks. The closest family of related work to the submission is the hypernetworks and the method neither discusses nor compares with them. This issue has been partially addressed in rebuttal with additional discussion and a small experiment on MultiMNIST; however, paper still needs a major revision of including hypernetworks as baseline in all experiments and rewriting related work and introduction sections by carefully positioning the paper with respect to hypernetworks. It is clear to me that authors need longer time to complete this revision and should submit to the next conference deadline.